# GISD30: global 30-m impervious surface dynamic dataset from 1985 to 2020 using time-series Landsat imagery on the Google Earth Engine platform

Xiao Zhang[1], Liangyun Liu[1, 2, *], Tingting Zhao[1, 3], Yuan Gao[1], Xidong Chen[1], Jun Mi[1, 2]

1 State Key Laboratory of Remote Sensing Science, Aerospace Information Research Institute, Chinese Academy of Sciences, Beijing 100094, China

2 University of Chinese Academy of Sciences, Beijing 100049, China

3 College of Geomatics, Xi'an University of Science and Technology, Xi'an 710054, China

* Corresponding author Email: liuly@radi.ac.cn

## Abstract

Accurately mapping impervious surface dynamics has great scientific significance and application value for urban sustainable development research, anthropogenic carbon emission assessment and global ecological environment modeling. In this study, a novel and automatic method by combining the advantages of spectral generalization and automatic sample extraction strategy was proposed and then an accurate global 30 m impervious surface dynamic dataset (GISD30) for 1985 to 2020 was produced using time-series Landsat imagery on the Google Earth Engine cloud-computing platform. Firstly, the global training samples and corresponding reflectance spectra were automatically derived from prior global 30 m land-cover products after employing the multitemporal compositing method and relative radiometric normalization. Then, spatiotemporal adaptive classification models, trained with the migrated reflectance spectra of impervious surfaces from 2020 and transferred pervious surface samples in each epoch for every 5°×5° geographical tile, were applied to map the impervious surface in each period. Furthermore, a spatiotemporal consistency correction method was presented to minimize the effects of independent classification errors and improve the spatiotemporal consistency of impervious surface dynamics. Our global 30 m impervious surface dynamic model achieved an overall accuracy of 90.1% and a kappa coefficient of 0.865 using 23,322 global time-series validation samples. Cross-comparisons with five existing global 30 m impervious surface products further indicated that our GISD30 dynamic product achieved the best performance in capturing the spatial distributions and spatiotemporal dynamics of impervious surfaces in various impervious landscapes. The statistical results indicated that the global impervious surface has doubled in the past 35 years, from $5.116\times10^5$ km² in 1985 to $10.871\times10^5$ km² in 2020, and Asia saw the largest increase in impervious surface area compared to other continents, with a total increase of $2.946\times10^5$ km². Therefore, it was concluded that our global 30 m impervious surface dynamic dataset is an accurate and promising product, and could provide vital support in monitoring regional or global urbanization as well as in related applications. The global 30 m impervious surface dynamic dataset from 1985 to 2020 generated in this paper is free to access at http://doi.org/10.5281/zenodo.5220816 (Liu et al., 2021b).

## 1. Introduction

Impervious surfaces are usually defined as surfaces "preventing the surface water from penetrating into the ground" and are composed of anthropogenic materials, such as steel, cement, asphalt, bricks and stone (Chen et al., 2016; Weng, 2012; Zhang et al., 2020). Over the past few decades, with the rapid growth of the population and the economy, impervious surfaces have been undergoing dramatic expansion, especially in developing countries (Gong et al., 2019a; Kuang, 2020). Based on the statistics of the United Nations in 2018, 55% of the

world's total population lives in cities, and this proportion is expected to reach 68% in 2050 (Unite Nations, 2019). As an indicator of the intensity of human activities and economic development, the dynamic information of impervious surfaces plays a significant role in urban planning (Li et al., 2015), biogeochemical cycles (Zhang and Weng, 2016), greenhouse gas emissions and urban heat island effects (Gao et al., 2012; Zhou et al., 2018), and urban sustainable development pathways (Liu et al., 2020b). Therefore, understanding and quantifying global impervious surface spatiotemporal dynamics is critical.

In recent years, with the continuous improvement of remote sensing techniques as well as computer storage and computing capabilities, global impervious surface monitoring has been undergoing a transition from the coarse spatial resolution of 1 km to the fine resolution of 30/10 m (Corbane et al., 2020; Gong et al., 2020; Liu et al., 2018; Liu et al., 2020b; Schneider et al., 2009; Zhao et al., 2020; Zhou et al., 2018). Specifically, coarse impervious surface products primarily use time-series nighttime light datasets (including DMSP and VIIRS NTL imagery) (Xie and Weng, 2017; Zhao et al., 2020) and MODIS imagery (Huang et al., 2020; Schneider et al., 2010) to capture global impervious surface dynamics; for example, Huang et al. (2021) used a fully automated mapping method to produce global 250 m urban area products for 2001 to 2018 using time-series MODIS imagery. Zhou et al. (2018) used the Defense Meteorological Satellite Program Linescane System's nighttime light data to develop temporally and spatially consistent global 1 km urban maps for 1992 to 2013. Although these coarse global impervious surface dynamic products could capture global urban expansion trends, they are unsuitable for many regional applications, because a large quantity of broken and small-sized impervious surfaces are missed in coarse remote sensing imagery (Gong et al., 2020). Recently, benefiting from the improvements and maturity of cloud computing platforms (such as Google Earth Engine (Gorelick et al., 2017)), many global 30 m multitemporal impervious surface products have been produced using long time-series Landsat imagery (Florczyk et al., 2019; Gong et al., 2020; Liu et al., 2018; Liu et al., 2020b). Liu et al. (2021a) comprehensively reviewed current seven global 30 m impervious surface products, and found only four products could capture the impervious expansion at the long time-series. Specifically, Liu et al. (2018) proposed a new index to develop multitemporal global 30 m urban land maps for 1990 to 2010 with 5-years intervals, but the products suffered the low producer's accuracy and user's accuracy of 0.50–0.60 and 0.49–0.61. Gong et al. (2020) used a combination of "exclusion–inclusion" and "temporal check" methods to generate the first annual global 30 m artificial impervious surface area dataset for 1985 to 2018, but the cross-comparisons in the Zhang et al. (2020) found that this annual dataset achieved great performance on mega-cities but suffered the under-estimation problems in the rural areas. The global human settlement layer (GHSL) monitored the impervious dynamic from 1975 to 2015 (Florczyk et al., 2019), but it suffered the overestimation problems at early stage and also missed the fragmented impervious objects (Gong et al., 2020). Therefore, an accurate global 30 m impervious surface dynamic product, which could accurately capture the spatiotemporal dynamic of various impervious objects including cities and rural, is still urgently needed.

Over the past few decades, many methods have been proposed for generating regional or global multitemporal impervious surface products. Generally, these methods can be divided into two groups: time-series change detection (Jing et al., 2021; Li et al., 2018; Song et al., 2016) and multitemporal independent classification/extraction (Gong et al., 2020; Liu et al., 2020b; Zhang and Weng, 2016). The time-series change detection strategy used change detection models to determine the break points in continuous Landsat observations. As this strategy makes full use of the correlations inherent within time-series imagery, it has higher robustness and a greater ability to capture urbanization time and frequency (Liu et al., 2019). However, as

impervious surfaces are usually nonlinear, with high temporal and spatial heterogeneity, impervious surface monitoring is a highly difficult and challenging task, especially for arid or semi-arid areas (Reba and Seto, 2020; Sexton et al., 2013). Zhu et al. (2019) demonstrated that the newest continuous monitoring of land disturbance (COLD) method still suffer from an omission error of 27% and a commission error of 28%. Meanwhile, the monitoring efficiency of the time-series change detection strategy is very low, because it uses pixel-by-pixel modeling by using continuous Landsat imagery.

The multitemporal independent classification/extraction strategy generates multiple temporally independent impervious surface maps, and then derives "from–to" information through per-pixel comparison, so the means of generating multiple temporally independent impervious surface maps is the key issue of the strategy. Zhang et al. (2020) concluded that there are three ways to generate independent impervious surface maps including: spectral mixture analysis (Wu, 2004; Zhuo et al., 2018), the spectral index-based method (Gao et al., 2012; Liu et al., 2018) and the image classification method (Zhang and Weng, 2016; Zhang et al., 2021a; Zhang et al., 2020). However, the spectral mixture analysis had great difficulty in finding the optimal endmembers, especially for long time-series monitoring. The spectral index-based method was simpler and more efficient than the other two strategies, but it encountered great difficulty in identifying the optimal threshold for deriving the impervious pixels from pervious surfaces, especially in arid areas (Sun et al., 2019). The image classification strategy uses training samples to build the classifiers for identifying impervious surfaces, and performed well in complex impervious surface mapping (Okujeni et al., 2013; Zhang et al., 2020). However, collecting training samples is a time-consuming and labor-intensive task, especially for large-area time-series impervious surface monitoring.

To solve the time-consuming and manual participation problems for collecting massive training samples, many studies have proposed to derive training samples from existing land-cover products after using a series of refinement rules, and successfully produced the large-area land-cover maps with fulfilling performances (Zhang and Roy, 2017; Zhang et al., 2021b; Zhang et al., 2019). For example, Zhang and Roy (2017) derived the training samples from time-series MCD12Q1 land-cover products and then used the derived samples for generating the 30 m land-cover maps with the overall accuracy of 95.44% over the whole America. Similarly, Zhang et al. (2021b) combined the CCI_LC land-cover products and time-series MCD43A4 to extract the confidence training samples and then produced the global 30 m land-cover products with the overall accuracy of 82.5%. However, it should be noted that the derived samples usually selected these spatiotemporal stable pixels as candidate samples for ensuring the confidence of training samples. Namely, these changed information cannot be captured using this derived strategy. In addition, the spectral generalization strategy had also been demonstrated to have great performance for automatic land-cover mapping (Phalke and Özdoğan, 2018; Wessels et al., 2016; Woodcock et al., 2001; Zhang et al., 2019). For example, Zhang et al. (2019) used the training spectra from MCD43A4 products to classify the multitemporal Landsat imagery in China with the overall accuracy of 80.7%. However, the spectral generalization strategy usually needed the prior reference training spectra to build the generalized classifier.

Monitoring impervious surface dynamics is a challenging and time-consuming task due to its high spatiotemporal heterogeneity. In this study, we proposed to a novel and automatic method by combining the advantages of spectral generalization and automatic sample extraction strategy for monitoring time-series impervious surface dynamics. Specifically, we derived the training samples from prior land-cover products to solve the time-consuming and manual participation problems for manually collecting massive training samples. Then, we combined the derived training samples and the temporally spectral generalization to independently

mapping impervious surfaces at long time-series. Next, a spatiotemporal consistency correction method was applied to the independent impervious surface maps to minimize the effects of classification errors and ensure the spatiotemporal consistency of the final dynamic impervious surface dataset. Finally, we produced an accurate and novel global 30 m impervious surface dynamic dataset (GISD30) from 1985 to 2020 by combining the proposed method and Google Earth Engine cloud computing platform, which also provide vital support for monitoring regional or global urbanization and performing related tasks.

## 2. Datasets

### 2.1 Time-series Landsat imagery

As a single Landsat mission cannot cover the whole period of 1985 to 2020 (Roy et al., 2014), all available Landsat imagery, including Landsat 4, 5, 7 and 8, archived on the GEE computation platform, were collected to monitor the spatiotemporal dynamics of impervious surfaces. To minimize the scattering and absorption effects of the atmosphere, all Landsat imagery was corrected for the surface reflectance using the Land Surface Reflectance Code (LaSRC) (Vermote et al., 2016) and Landsat Ecosystem Disturbance Adaptive Processing System (LEDAPS) (Vermote, 2007) algorithms. Meanwhile, these poor observations (including snow, shadow, cloud and saturated pixels) in the Landsat imagery were masked using the CFmask algorithm (Zhu and Woodcock, 2014), which is the official Landsat processing algorithm and is included in the Landsat Surface Reflectance (SR) Product Handbook (USGS, 2017). Figure 1 illustrates the spatial distributions of all available Landsat observations from 1985 to 2020, with intervals of 5-years; clearly, the availability of Landsat imagery had a significant positive relationship with the advancement of the monitoring period, mainly because later Landsat satellites had greater capacities for onboard recording and satellite-to-ground transmission compared with previous Landsat systems (Roy et al., 2014). In addition, as only Landsat 5 could provide observation imagery, and satellite-to-ground transmission capabilities were fairly low before 2000, the available Landsat observations before 2000 cannot cover the whole world, and those for 1985 are especially limited; however, it should be noted that we assumed that the land-cover in these areas with missing data would remain stable during the period.

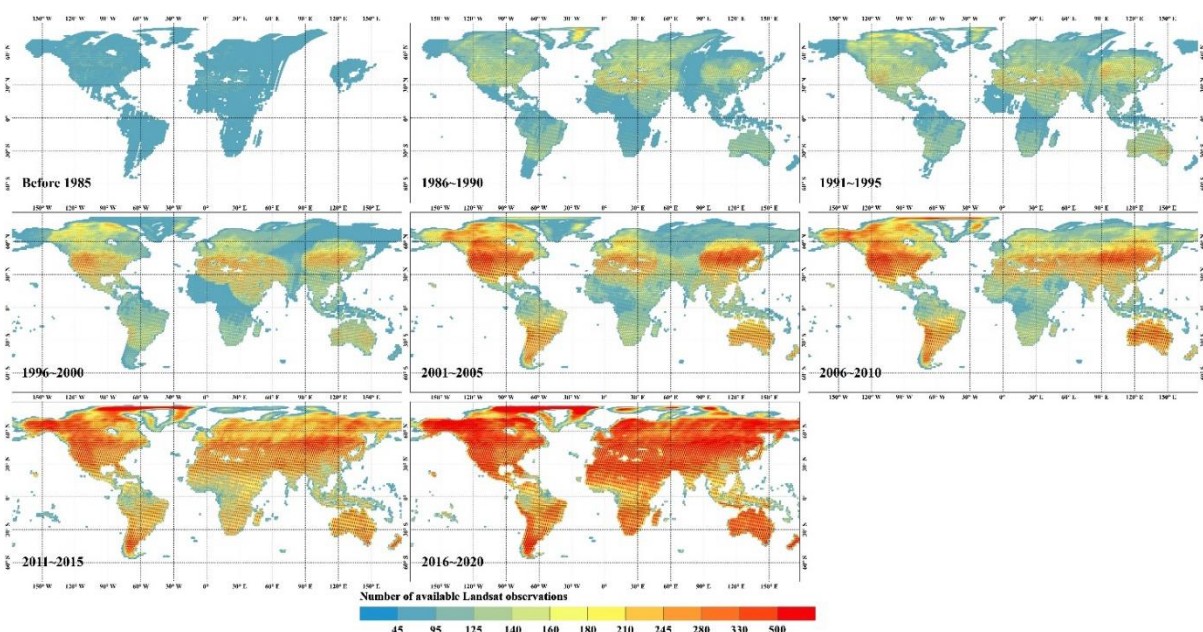

Figure 1. The spatial distributions of the available Landsat observations from 1985 to 2020 with 5-year intervals.

## 2.2 Global 30 m land-cover product in 2020

To automatically monitor the spatiotemporal dynamics of impervious surfaces, it was necessary to import a global 30 m land-cover product from 2020, which was used as the reference dataset for deriving training samples in Section 3.1, and provided the broadest impervious surface extents for monitoring spatiotemporal dynamics. In this study, the GLC_FCS30-2020 (Global Land Cover product with Fine Classification System at 30 m in 2020) dataset, generated by combining the time-series of Landsat imagery with high-quality training data from the Global Spatial Temporal Spectra Library on the Google Earth Engine computing platform (Zhang et al., 2021b), was used, showing an overall accuracy of 82.5% and a kappa coefficient of 0.784 for the level-0 validation system (9 basic land-cover types), and an overall accuracy of 68.7% and kappa coefficient of 0.662 for the UN-LCCS level-2 system (24 fine land-cover types), employing 44,043 global validation samples (Zhang et al., 2021b). It should be noted that the impervious surface layer in the GLC_FCS30-2020 dataset, which was independently produced by combining multisource and multitemporal remote sensing imagery and achieved an overall accuracy of 95.1% and a kappa coefficient of 0.898 (Zhang et al., 2020), was not used as the result for period of 2015-2020 in the final results, instead, only used as the prior dataset for deriving training samples and determining the broadest extents. The GLC_FCS30-2020 dataset is free to access at http://doi.org/10.5281/zenodo.4280923 (Liu et al., 2020a).

## 2.3 Validation dataset

To quantitatively assess the accuracies of our impervious surface dynamic time-series products, 23,322 validation samples (Figure 2), including 13,336 impervious samples and 9,986 pervious samples, covering the long-term time-series from 1985 to 2020, were randomly generated using the stratified random sampling strategy, and further interpreted on the Google Earth Engine computing platform. The GEE computing platform had obvious advantages over collecting validation samples, including: 1) storing massive amounts of remote sensing imagery with various spatial resolutions and time spans; 2) easy access to different remote sensing images via simplified coding (Gorelick et al., 2017). Therefore, using multisource high-resolution imagery archived in the GEE platform, each validation sample could be marked as "pervious surface" or "specific change year of impervious surface". However, as the high-resolution images from 1985 to 2000 were sparse, and the Landsat imagery contained observations for that period with satisfactory spatial resolution, we used the time-series Landsat imagery as the auxiliary dataset for visual interpretation between 1985 and 2000. Further, as the spatial heterogeneity of the impervious surface was usually higher than that of natural land-cover types, the impervious area in a $30 \times 30$ m window should comprise more than 50% when identifying impervious samples (Zhang et al., 2020). Meanwhile, to minimize the effect of geometry registration between validation samples and our products, the geolocations of these rural impervious surface samples, located in the transition areas of the impervious objects (such as buildings and roads) and pervious surfaces, were re-positioned in the center of the objects. Lastly, to minimize the influence of the interpreting experts' subjective knowledge, each validation sample was to be independently interpreted by five experts.

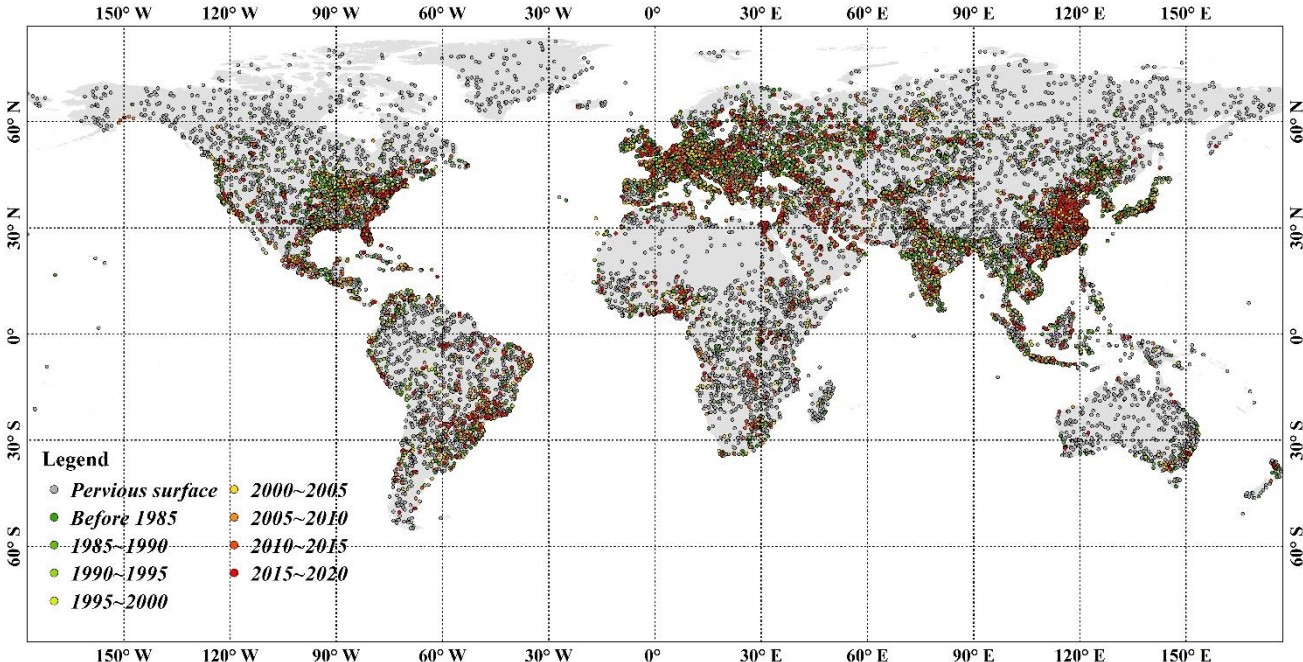

Figure 2. The spatial distribution of the global multitemporal impervious surface validation dataset for 1985-2020.

## 2.4 Existing multitemporal global 30 m impervious surface products

In this study, five existing multitemporal global 30 m impervious surface products, including GAIA (Global Artificial Impervious Area), GHSL (Global Human Settlement Layer), GAUD (Global Annual Urban Dynamics), GlobeLand30 impervious surface layer and NUACI (Normalized Urban Areas Composite Index) - based maps, were used to comprehensively analyze the performance of our products. Specifically, GAIA was generated by combining the "Exclusion/Inclusion" and "Temporal Consistency" methods and applying them to time-series Landsat imagery, which provided the global annual impervious surface from 1985 to 2018 at a 30 m spatial resolution, with a mean accuracy of 90% using 3500 validation samples (Gong et al., 2020). The GHSL products, developed by fusing supervised and unsupervised classification processes to achieve a combination of data-driven and knowledge-driven processes, contained four epochs' impervious surface dynamics (1970, 1990, 2000 and 2015) (Florczyk et al., 2019; Pesaresi et al., 2016), with the high overall accuracy of 96.28% and the low kappa coefficient of 0.323 using the open LUCAS (Land-Use/Cover Area Frame Survey) validation dataset for Europe (Pesaresi et al., 2016). The GAUD dataset, produced by combining four prior global urban-extent maps and time-series normalized urban areas composite index, monitored annual changes in urban extent from 1985 to 2015 and achieved the mean kappa coefficient of 0.57 in 2015 (Liu et al., 2020b). The GlobeLand30 impervious surface layer, which was an independent land-cover type in the GlobeLand30 global land-cover product, was produced by combining pixel-based classification, multi-scale object-oriented segmentation and manual verification based on the visual interpretation of high-spatial resolution imagery (Chen et al., 2015). Meanwhile, to eliminate salt and pepper noise in the impervious surface layer, a minimum unit of 4 ×4 pixels was applied for each impervious surface object. In this study, three epochs' (2000, 2010 and 2020) impervious surface layers were included in the GlobeLand30, and independent validation indicated that the accuracy of impervious surface was over 80% (Chen and Chen, 2018; Chen et al., 2016). The NUACI-based products were generated by combining the multi-temporal NUACI index and adaptive threshold

optimization methods and applying them to the time-series Landsat and nighttime light imagery (Liu et al., 2018), which contained the impervious surface dynamics of seven epochs from 1985 to 2015 with five-year intervals. The independent validation indicated that the NUACI-based products achieved overall accuracy, producer's accuracy and user's accuracy of 0.81–0.84, 0.50–0.60 and 0.49–0.61, respectively (Liu et al., 2018).

## 3. Methods

In this study, a novel and automatic method, combining temporally spectral generalization and automatic training sample extraction strategies, was proposed to automatically monitor the spatiotemporal dynamics of impervious surfaces. Specifically, the training samples and maximum impervious surface extents in 2020 were firstly derived from the prior GLC_FCS30-2020 land-cover products and other global 30 m impervious surface products. Secondly, based on the assumption that the land-cover transition from impervious surface to pervious surface was irreversible, the pervious surface samples in 2020 were directly transferred to other periods. As for the impervious surface samples, as it was impossible to directly transform them, we proposed to migrate their reflectance spectra in 2020 to other periods by using the temporally spectral generalization strategy. Thirdly, multitemporal local adaptive random forest classification models, trained by the migrated reflectance spectra of impervious surfaces in 2020 and transferred pervious surface samples, were applied to independently generate time-series impervious surface maps from 1985 to 2020. Lastly, the temporal consistency checking method was used to ensure the spatiotemporal consistency and logic of using this approach for monitoring the spatiotemporal dynamics of impervious surfaces.

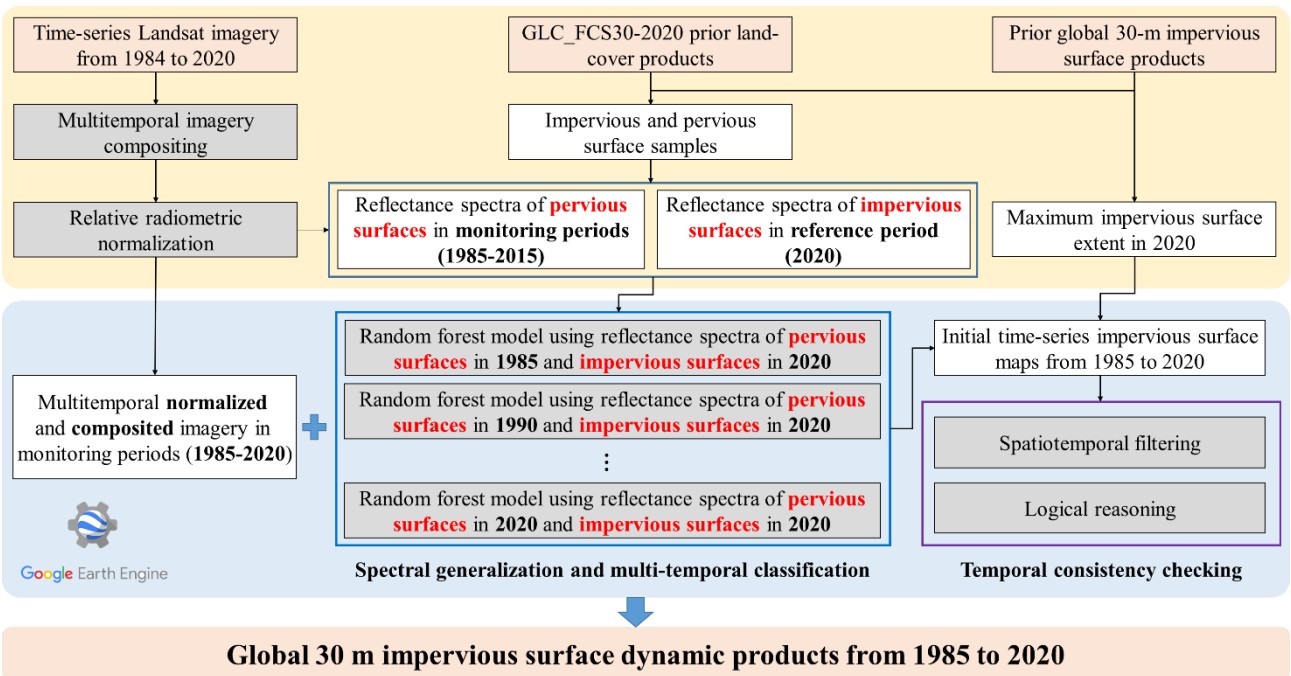

Figure 3. The flowchart of the spectral generalization method for automatically monitoring the spatiotemporal dynamics of impervious surface from 1985 to 2020.

## 3.1 Deriving training reflectance spectra and maximum impervious surface boundary

To achieve the automatic monitoring of the spatiotemporal dynamics of impervious surfaces, we proposed to transfer the pervious samples in 2020 to other periods by the irreversible assumption, and simultaneously migrated the impervious reflectance spectra in 2020 to other periods using spectral generalization strategy. The

key steps of the novel method were: 1) deriving training samples and maximum impervious surface extents from the prior GLC_FCS30-2020 land-cover products and other impervious surface products; 2) multitemporal imagery compositing and relative radiometric normalization, which guarantees the feasibility of migrating the reflectance spectra of impervious surfaces in 2020 to other periods.

### 3.1.1 Deriving training samples and maximum impervious surface extents from existing products

As opposed to the traditional method of collecting training samples based on visual interpretation, in this study, the global training samples, including those of the impervious surface and the pervious surfaces, were automatically derived from the prior GLC_FCS30-2020 land-cover products and other prior impervious surface products by using a series of refinement rules. The reasons why we mainly chose the GLC_FCS30-2020 as the reference dataset were because: 1) the impervious surface layer in the GLC_FCS30-2020 was independently

produced by combining multisource and multitemporal imagery with the high user's accuracy of 93.2% and a producer's accuracy of 94.8% (Zhang et al., 2020); 2) the other pervious land-cover types in the GLC_FCS30-2020 also achieved a great performance with the overall accuracy 82.5%. Specifically, we firstly determined the maximum impervious surface extents and impervious training samples from several prior products. Although the impervious layer in GLC_FCS30-2020 had an omission error of only 5.2% (Zhang et al., 2020),

we still combined multiple global 30 m impervious surface products (GAIA-2018 ($IS_{gaia}$), GHSL-2014 ($IS_{ghsl}$), impervious layer in the GlobeLand30-2020 ($IS_{globleand30}$) and GLC_FCS30-2020 ($IS_{glc\_fcs30}$)) to capture all the impervious surfaces as comprehensive as possible. Namely, the maximum impervious surface extents ($IS_{max}$), derived via the union of these four global impervious surface products (formula (1)), was used as the maximum boundary of subsequent time-series classifications.

$$IS_{max} = IS_{gaia} \cup IS_{ghsl} \cup IS_{globleand30} \cup IS_{glc\_fcs30} \qquad (1)$$

    Then, as for how to derive impervious training samples, the GAIA and GHSL datasets were demonstrated to suffer the problem of missing these fragmented impervious objects (such as: rural villages, roads) (Sun et al., 2019), so the intersection operation was only applied to the impervious layer in the GlobeLand30-2020 ($IS_{globleand30}$) and GLC_FCS30-2020 ($IS_{glc\_fcs30}$) to comprehensively capture impervious samples in both

cities or small villages and minimize the effect of commission error in this two products. Afterwards, as the transition areas between two different land-cover types had high probability of being misclassified (Radoux et al., 2014), the spatial homogeneity of each candidate impervious sample was calculated using a local window of 3×3:

$$P_{x,y} = \frac{1}{N}\left[\sum_{x-1}^{x+1}\sum_{y-1}^{y+1} I\left(L_{x,y} = L_{imp}\right)\right] \qquad (2)$$

where $P_{x,y}$ denotes the spatial homogeneity of candidate pixel $L_{x,y}$, $L_{imp}$ represents the label value of impervious surface, and the $I\left(L_{x,y} = L_{imp}\right)$ is the indicator function and N is the size of the local window size. In this study, we only retained these spatial homogeneity candidate impervious samples. Namely, if the $P_{x,y}$ of candidate pixel was less than 1, the candidate impervious sample would be discarded.

    As we have combined four prior 30 m impervious surface products to determine the maximum impervious

surface extents in 2020 ($IS_{max}$), the remaining areas outside $IS_{max}$ were considered as pervious surfaces ($PS_{condi}$). However, due to the complicated makeup and spectral heterogeneity of impervious surfaces, some pervious surface types such as: bare land, grassland and cropland would be spectrally confused with the impervious surfaces. For example, bare land was spectrally similar to the high-reflectance impervious surfaces

because composition materials of the impervious surface, including the cement bricks and stone, were also present in the bare land. Meanwhile, cropland was also easily confused with impervious surfaces, especially in the cases of some rural buildings (Sun et al., 2019), because both are composed of low-reflectance vegetation and high-reflectance artificial materials or bare soil. Therefore, we proposed to further split $PS_{candi}$ into three sub-categories (cropland, bare land and others) by using the GLC_FCS30-2020. Meanwhile, the spatial homogeneity checking (formula (2)) was also applied to each $PS_{condi}$ sample to minimize the confusions in these land-cover transition areas.

Although we used refinement rules to extract high confidence training samples, the volume of candidate training points (including impervious surface and pervious surfaces) was still large especially for the pervious samples. Some studies have quantitatively demonstrated that the distribution, balance and size of training samples affect the classification accuracy (Jin et al., 2014; Mellor et al., 2015; Zhu et al., 2016). In this study, as the impervious surface was sparser than other pervious surfaces (cropland, bare land and others) in term of global total distribution areas, the training samples with equal allocation were used to guarantee training sample balance and to capture the spectral heterogeneity of impervious surfaces as effectively as possible. Namely, the ratio of impervious samples and pervious samples was close to 1:3. In addition, the spatial distribution of impervious surfaces greatly varies in different regions, therefore, if we derived training samples on a global scale, the continents with more sparse impervious surfaces (South America, Africa and Oceania) would lack sufficient samples to characterize their impervious surfaces. In order to further ensure that the training samples were locally adaptive, we adopted the tiled solution used in (Zhang et al., 2021b), splitting the global land-area into approximately 961 5°×5° geographical tiles (Figure 4), and independently deriving training samples for each geographical tile. As for the sample size in each tile, Zhu et al. (2016) quantitatively demonstrated that the mapping accuracy first increased and then stabilized with the increase of the sample size and suggested a minimum of 600 training samples and a maximum of 8000 training samples per class. In this study, the sample size was about 5000 for each class, and the ratio between impervious surfaces and pervious surfaces was 1:3.

### 3.1.2 Multitemporal imagery composting and relative radiometric normalization

As our previous work (Zhang et al. (2020)) had quantitatively demonstrated that multitemporal information made a positive contribution to large-area impervious surface mapping, and the availability of Landsat imagery varied with the spatial distribution in Figure 1, it was necessary to decompose the time-series Landsat imagery into multitemporal features. According to the reviews in the works of Gomez et al. (2016), there were two main options—"selection-based" and "transform-based"—for extracting multitemporal information from time-series imagery. The "selection-based" option was to use user-defined criteria to select the most suitable observation from the time-series imagery, so the composited imagery still contained the characteristics of surface reflectance. For example, the maximum NDVI (Normalized Difference Vegetation Index) compositing method was to select the observation with the largest NDVI value from time-series observations. While the "transform-based" method was to use the transform models (Fourier transform, mathematical statistics, etc.) to transform the time-series observations into new variables band by band, for example, the widely used quantile compositing method was to transform the time-series spectra into several quantiles based on the ranking of the values. Therefore, the composited imagery derived by the "transform-based" strategy cannot represented the actual characteristics of surface reflectance at wavelength dimension.

In this study, as we needed to migrate the reflectance spectra of impervious surfaces in 2020 to other periods, the "selection-based" strategy was the optimal solution for spectral generalization. To select the user-defined

criteria to composite the multitemporal features, given that the best-available-pixel (BAP) method could simultaneously take into account four factors (sensor type, day of year, distance to cloud or cloud shadow and aerosol optical thickness (White et al., 2014)), it has been widely used for generating annual or seasonal cloud-free composited imagery (Chen et al., 2021; Liu et al., 2019). In this study, in order to capture the multitemporal information from the time-series Landsat imagery, the seasonal BAP composited method, which applied the BAP compositing approach for each season, was used on time-series Landsat imagery in each period. Therefore, we derived four sets of seasonally composited Landsat imagery for each period. It should be noted that we categorized the time-series Landsat imagery from 1982 to 2020 into 8 periods with the interval of 5 years corresponding to the Figure 1. Meanwhile, we also assumed that the land-cover in those no Landsat observation areas would remain stable during the period. According to our statistics, the missing Landsat observations during 1986-1995 mainly concentrated on the Northeast Asia in which contained a small number of impervious surfaces.

Meanwhile, for each seasonally composited imagery, excluding those in six optical bands (blue, green, red, NIR, SWIR1 and SWIR2), three spectral indexes, including the normalized difference built-up index (NDBI), normalized difference water index (NDWI) and normalized difference vegetation index (NDVI), were also imported, because NDBI was a good indicator of impervious surface and bareland, NDVI was sensitive to the vegetation, and NDWI was one of the most popular indices for mapping water bodies. Eventually, a total of 36 multitemporal spectral features were derived for four seasonal composites.

$$NDBI = \frac{\rho_{swir1} - \rho_{nir}}{\rho_{swir1} + \rho_{nir}}, \; NDWI = \frac{\rho_{green} - \rho_{swir1}}{\rho_{green} + \rho_{swir1}}, \; NDVI = \frac{\rho_{nir} - \rho_{red}}{\rho_{nir} + \rho_{red}} \qquad (3)$$

Afterwards, the prerequisite for temporally spectral generalization was the spectral consistency between reference imagery and unclassified imagery. In this study, some measures were taken to ensure the highest possible spectral consistency in the Landsat composited imagery for the reference period and other periods: 1) the "selection-based" strategy was applied to ensure that the composited imagery could characterize the reflective characteristics of the land surface; 2) the seasonal BAP method was used to guarantee the phenological consistency of each set of seasonally composited imagery. However, there was still a small difference in the spectral response between Landsat sensors (TM, ETM+ and OLI) (Roy et al., 2016), and some factors (including the number of available Landsat observations, frequency of cloud and shadow, etc.) caused small temporal difference in the seasonal composites between the reference imagery and unclassified imagery. Therefore, we used the relative radiometric normalization method to further ensure the spectral consistency between reference and unclassified imagery. Specifically, as we migrated the reflectance spectra of impervious surfaces in 2020 to other periods, the seasonal composites in 2020 were the dependent variables ($\rho_{R,S_j}(\lambda_i)$):

$$\rho_{R,S_j}(\lambda_i) = \alpha_i \times \rho_{t,S_j}(\lambda_i) + \beta_i \qquad (4)$$

where $\rho_{t,S_j}(\lambda_i)$ was the surface reflectance in band $\lambda_i$ in the period $t$ ($t = 1985, 1990, \ldots, 2015$), $S_j$ represented the seasonal composites in different seasons, and $\alpha_i$ and $\beta_i$ denoted the slope and intercept of the linear regression model.

## 3.2 Spectral generalization classification and temporal consistency checking

Based on the assumption that the land-cover transition from impervious surface to pervious surface was irreversible, the derived pervious samples in 2020 (Section 3.1.1) would be directly transferred to other periods, but the impervious surface samples in 2020 cannot be transferred. To solve the lack of impervious surface

samples before 2020, we normalized the reflectance spectra of impervious surfaces in other epochs to those in
2020 using the relative radiometric normalization method (Section 3.1.2). Specifically, we independently
trained the classification models at each period using the generalized impervious reflectance spectra
($TrainFeatures\_IS_{2020}$) and the pervious samples ($TrainFeatures\_PS_t$) as:

$$TrainFeatures\_PS_t = \left[ \sum_{s_i} \left( \rho_b^{s_i,t}, \rho_g^{s_i,t}, \rho_r^{s_i,t}, \rho_{nir}^{s_i,t}, \rho_{swir1}^{s_i,t}, \rho_{swir2}^{s_i,t}, ndbi^{s_i,t}, ndvi^{s_i,t}, ndwi^{s_i,t} \right) \right]$$

$$TrainFeatures\_IS_{2020} = \left[ \sum_{s_i} \left( \rho_b^{s_i}, \rho_g^{s_i}, \rho_r^{s_i}, \rho_{nir}^{s_i}, \rho_{swir1}^{s_i}, \rho_{swir2}^{s_i}, ndbi^{s_i}, ndvi^{s_i}, ndwi^{s_i} \right) \right]$$

(5)

where $s_i$ denotes various seasonal composites and $t$ is the monitored period. It can be found that the
$TrainFeatures\_PS_t$ varies with the $t$, namely, the training spectra of pervious surfaces directly came from
the unclassified imagery. It should be noted that there may not be cloud-free imagery available especially for
the rainy season before 2000 in the tropical rainforest areas. In this case, we discarded this missed seasonal
features when training the classification models, namely, the number of training features varied with the
availability of Landsat observations.

Afterwards, as the spatial distributions of impervious surfaces varied in different regions, we used the local
adaptive resampling strategy to comprehensively capture the impervious surface characteristics at various
regions (Section 3.1.1). However, if we used all training samples to build a global classification model for
mapping global impervious surfaces, the global model still sacrifice the performance in these sparse impervious
surface regions to achieve high overall accuracy. In this study, the local adaptive modeling strategy, spited the
globe into multiple local regions and then independently trained the classification models in each local region
using corresponding regional samples, was adopted to increase the sensitivity and fitting ability of the
classification model at different regions. Zhang and Roy (2017) also quantitatively compared the performance
of global classification modelling and local adaptive modelling strategies and found the latter had greater
performance than the former. Therefore, we adopted the tiled solution used in the Zhang et al. (2021b), splitting
the global land-area into approximately 961 5°×5° geographical tiles (Figure 4), and then trained independent
classification model in each geographical tiles.

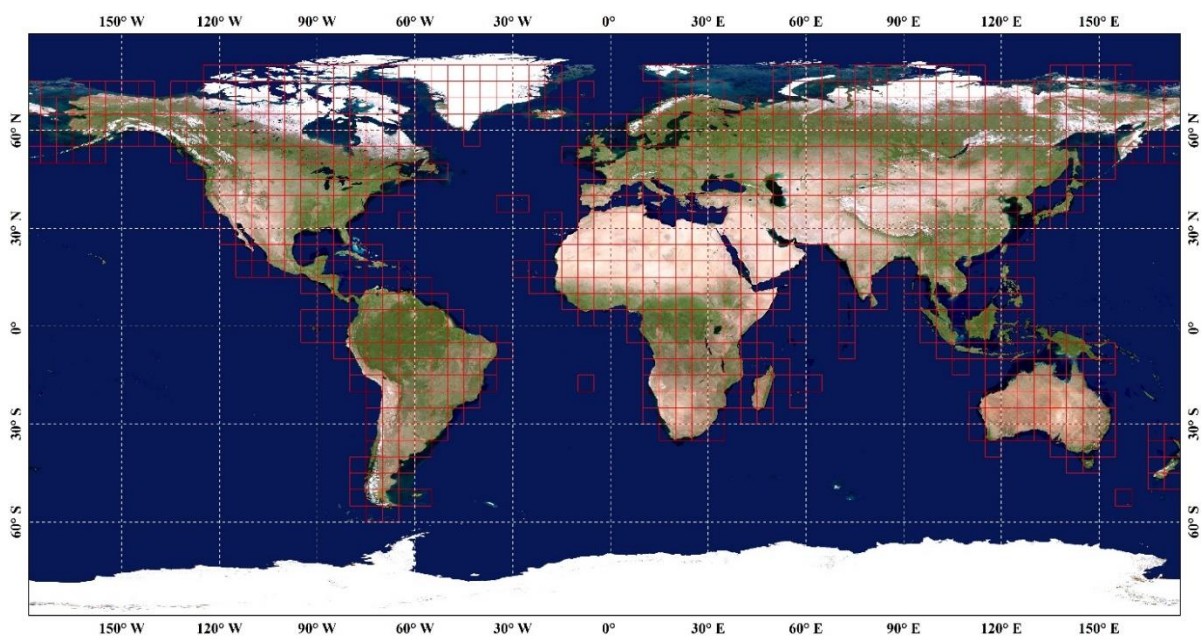

Figure 4. The spatial distribution of 961 5°×5° geographical tiles for local adaptive modeling. The background imagery comes from the National Aeronautics and Space Administration (https://visibleearth.nasa.gov).

Furthermore, the random forest (RF) classification model has significant advantages over other classification models (such as decision tree, support vector machine and neural network), including: 1) higher computation efficiency and classification accuracy; 2) a stronger ability to process high-dimensional data and resist training sample errors; 3) simpler parameter settings (Belgiu and Drăguţ, 2016; Du et al., 2015; Gislason et al., 2006). Therefore, the RF classifier was selected to produce our impervious surface dynamic time-series products. The RF classifier only contains two adjustable parameters (the number of decision trees (Ntree) and the number of selected prediction variables (Mtry)), and Belgiu and Drăguţ (2016) quantitatively analyzed the relationship between the classification accuracy and these two parameters, finding that the Ntree had a greater impact on classification accuracy than Mtry and suggesting that these two parameters should take default values. As such, we defined the Ntree as 500 and Mtry as the square root of the total number of input features.

Lastly, as the time-series impervious surface products were produced by independent classifications, it was necessary to use the post-processing method to optimize the time-series impervious products from 1985 to 2020 and minimize the influence of classification error. Over the past few years, many post-processing methods have been proposed, including maximum a posteriori Markov random fields (Cai et al., 2014) and temporal consistency checks (Li et al., 2015), both of which use contextual spatiotemporal information and prior knowledge to reduce the illogical land-cover transitions caused by classification error. In this study, the "temporal consistency correction" proposed by (Li et al., 2015) was applied to optimize our impervious time-series products. It mainly comprised procedures of spatiotemporal filtering and illogical transition checking, the former of which iteratively calculates the probability of the same land-cover pixels occurring in the neighborhoods within a 3×3×3 spatiotemporal window as:

$$P_{x,y,t} = \frac{1}{N}\left[\sum_{x'=x-1}^{x'=x+1}\sum_{y'=y-1}^{y'=y+1}\sum_{t'=t-1}^{t'=t+1} I\left(L_{x',y',t'} = L_{x,y,t}\right)\right] \qquad (6)$$

where $L_{x',y',t'}$ denotes the adjacent pixels in the spatiotemporal window, $L_{x,y,t}$ reprensents the label of the target pixel $(x, y)$ in the period of $t$, and $I\left(L_{x',y',t'} = L_{x,y,t}\right)$ is the indicator function. Usually, the value of $P_{x,y,t}$ could reflect the accuracy of $L_{x,y,t}$, namely, a higher value of $P_{x,y,t}$ means the high confidence of $L_{x,y,t}$. In this study, the threshold of 0.5 for the $P_{x,y,t}$ (suggested by the Li et al., 2015) was applied to reduce the influence of classification error caused by individual classifications. If the $P_{x,y,t}$ for each impervious surface pixel was lower than 0.5, the corresponding label was adjusted as the opposite. Afterwards, the illogical transition checking mainly employed the irreversibility assumption to remove illogical transitions from impervious surface to pervious surface.

## 3.3  Accuracy assessment

To comprehensively assess the performance of our global 30 m impervious surface dynamic dataset, sample-based and comparison-based methods were applied. Specifically, the sample-based validation method used the multitemporal impervious surface validation samples to calculate four accuracy metrics, including the overall accuracy and kappa coefficient, the producer's accuracy (measuring the commission error) and the user's accuracy (measuring the omission error) (Olofsson et al., 2014). Meanwhile, as opposed to traditional period-by-period accuracy assessments, we categorized the time-series impervious surface dynamic into 9 independent strata, including: pervious surfaces, impervious surfaces before 1985, and expanded impervious surfaces during

1985-1990, 1990-1995, 1995-2000, 2000-2005, 2005-2010, 2010-2015 and 2015-2020. We then calculated a comprehensive confusion matrix for these nine strata.

In addition, the comparison-based method used five global 30 m impervious surface products (GAIA, GHSL, NUACI, GAUD and GlobeLand30) with multiple epochs as the comparative dataset for analyzing the performance of our GISD30 products. Specifically, we compared the time-series impervious areas of five products in six continents, and further analyzed the spatial consistency between GISD30 and five comparative datasets at the global scale. Further, we selected three types of cities (mega-cities, tropical cities and arid cities) and one rural area to illustrate the performance of five global 30 m impervious surface products used for capturing the spatiotemporal dynamic. The reasons why we chose these types of cities and rural areas were that (1) the mega-cities usually experienced more intense urbanization, we could more intuitively understand whether there were commission error and omission error in each product; (2) the tropical cities usually mean sparser observations caused by the cloud coverage, so we could analyze the stability and robustness of each product in the tropical cities; (3) the arid cities were selected to analyze the ability of each product to distinguish between impervious surfaces and similar land types (arid soils); (4) the rural area contained sparse impervious surfaces and were prone to suffer the underestimation problem.

## 4    Results

### 4.1    The spatiotemporal dynamics of impervious surfaces from 1985 to 2020

Figure 5 illustrated the spatial distributions of time-series global 30 m impervious surface maps and two local enlargements in China and India during 1985-2020 with intervals of 5 years. Intuitively, as the world's main impervious surfaces and economic activities are mainly concentrated in the northern hemisphere, the intensity of impervious surface expansion in the northern hemisphere is more significant than that in the southern hemisphere. Specifically, the impervious surfaces have undergone rapid urbanization in past 35 years especially in developing countries such as China and India in Figure 5a and b. It can be found that many low-density areas in 1985 were transformed into medium/high-density areas in 2020, and the cities were obviously connected by the new impervious surfaces especially in the mega-cities such as Shanghai and Guangzhou in China. Meanwhile, the cities (such as: Bangkok, New Delhi and Beijing in Figure 5a and b) usually experienced faster impervious surface expansion speeds than the surrounding villages and small cities, etc.

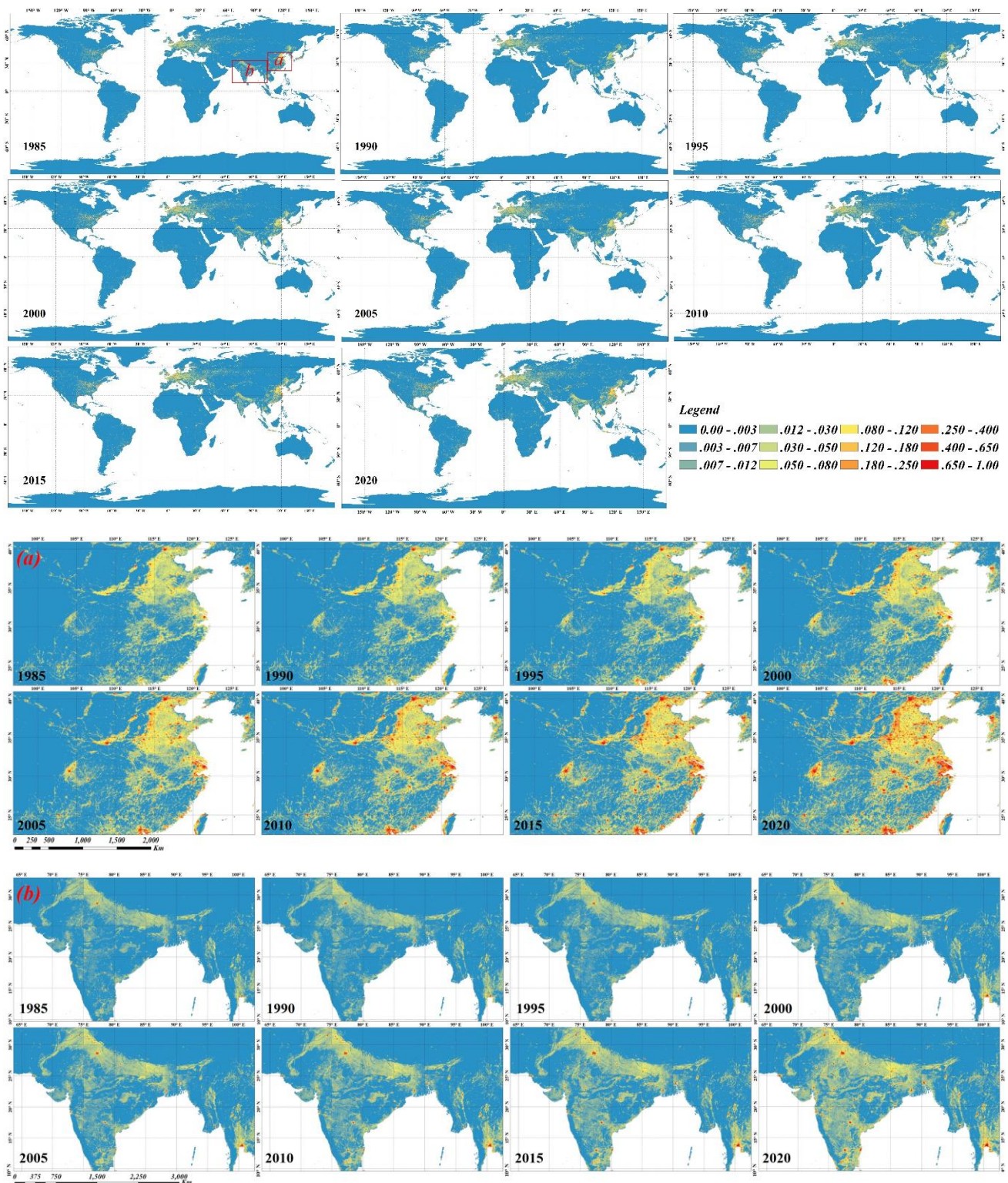

Figure 5. The spatial distributions of time-series global 30 m impervious surface results and two local enlargements in China and India from 1985 to 2020 with intervals of 5 years. Each pixel represents the fraction of impervious surface within each 0.05°×0.05° spatial grid.

Figure 6 quantitatively summarizes the impervious surface areas and their changes on six continents from 1985 to 2020. Overall, the global impervious surface area has doubled in the past 35 years, from $5.116\times10^5$ km$^2$ in 1985 to $10.871\times10^5$ km$^2$ in 2020. Specifically, Asia experienced the largest increase in impervious surface

area compared to other continents, with a total increase of $2.946 \times 10^5$ km² (from $1.908 \times 10^5$ km² in 1985 to $4.854 \times 10^5$ km² in 2020), followed by North America (from $1.202 \times 10^5$ km² to $2.188 \times 10^5$ km²), Europe (from $1.330 \times 10^5$ km² to $2.168 \times 10^5$ km²), Africa (from $0.264 \times 10^5$ km² to $0.725 \times 10^5$ km²), and South America (from $0.298 \times 10^5$ km² to $0.735 \times 10^5$ km²), and Oceania experienced lowest urbanization, with an increase of $0.088 \times 10^5$ km² over the past 35 years. In addition, Figure 6b indicated that the proportion of impervious area on Asia, Africa and South America continents obviously increased, while the proportions of the remaining three continents (Europe, North America and Oceania) slowly declined during 1985-2020. Specifically, the proportion of impervious area in Asia increased the most, from 37.3% to 44.7%, while the proportion in Europe clearly decreased, from 26.0% to 20.1%. Lastly, Figure 6d illustrates the impervious surface expansion ratio of six continents in 1985-2020. Africa displayed the fastest expansion ratio compared to other continents—the impervious area in Africa was 1.74 times greater than that in 1985, followed by Asia and South America, with expansion ratios of 154.4% and 146.4% over the period, respectively. Comparatively, as Europe and North America had large impervious surface areas in 1985, their impervious area expansion ratios were relatively low. Meanwhile, it can be found that the expansion rate of impervious surface area on six continents after 2000 was significantly faster than before 2000.

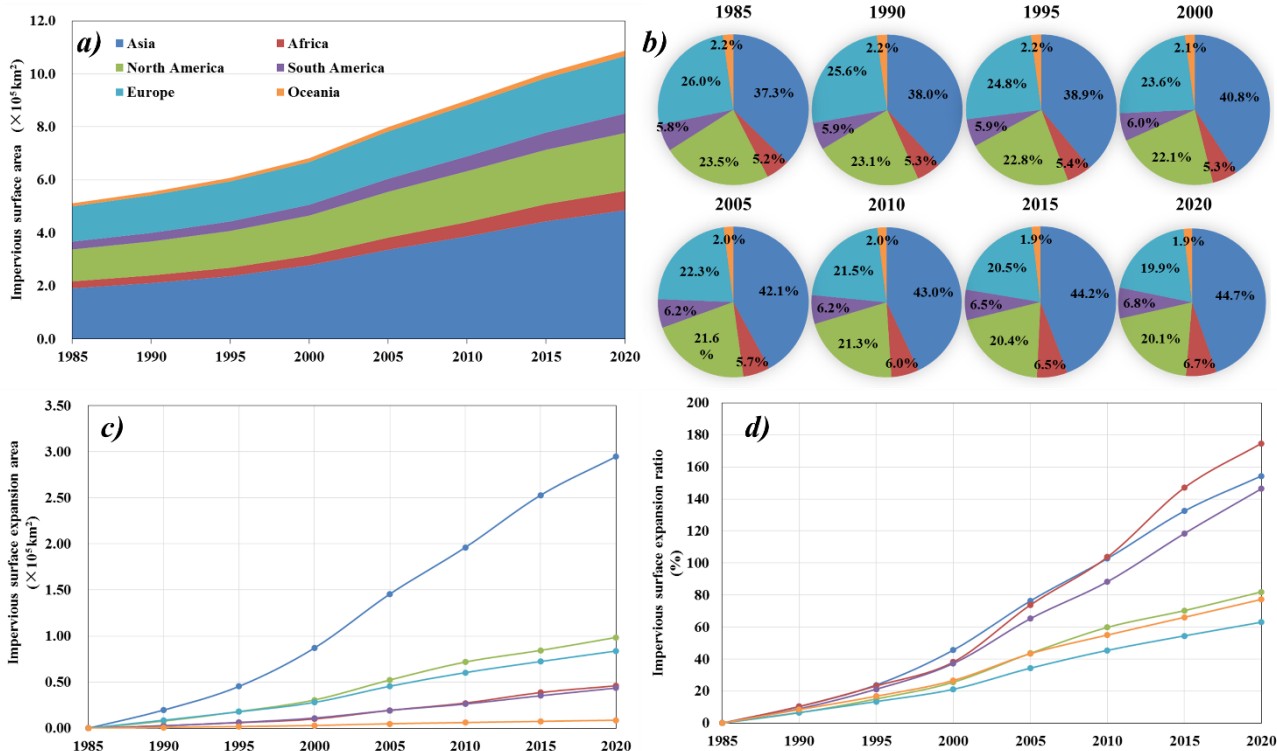

Figure 6. The expansion of impervious surfaces on each continent over the period of 1985-2020. (a) The impervious areas of six continents in each period. (b) The proportion of impervious areas on six continents from 1985 to 2020. (c-d) The increased impervious area and corresponding expansion ratio on each continent.

Figure 7 quantitatively measures the growth of impervious surfaces in various countries around the world over the period of 1985-2020. Intuitively, China underwent the largest increase in impervious area in the last 35 years, with an increase of $1.31 \times 10^5$ km², followed by America and India both exceeding $4.0 \times 10^4$ km², and Russia and Brazil exceeding $2.0 \times 10^4$ km². Meanwhile, from the perspective of spatial distribution, countries in Asia and North America displayed a higher increase in impervious area than those in other continents, especially East Asian and South Asian countries. In comparison, most countries in Africa underwent relatively little impervious

surface growth, with an increase of less than 4000 km² over the past 35 years. Although Europe is a center of global economic activity, the increased impervious area in European countries was not significant compared with North America and Southeast Asia, and the average increase in area was less than 8000 km². In addition,

Figure 7 shows the sum of the impervious surface area in the meridional and zonal directions in 1985 (blue) and 2020 (red), with a step of 0.05°, respectively. The meridional statistics indicated that the impervious surface in 1985 was more evenly distributed in the meridional direction than that in 2020. In 2020, there were four distinct peak intervals: 100°W~70°W (covering eastern United States), 0°~50°E (containing most European countries), 70°E~90°E (covering the whole of India) and 100°E~120°E (containing many Southeast Asia countries and

China). Meanwhile, the increase in impervious area in the Eastern Hemisphere was significantly larger than that in the Western Hemisphere, and the maximum increase in impervious area was located near 120°E, containing China's three major economic deltas (Yangtze River Delta, Pearl River Delta and Jing-Jin-Ji metropolitan region). Next, the zonal statistics indicated that the vast majority of impervious surfaces in the world were distributed between approximately 20°N and 60°N, the area of which contains most of the world's economically

developed and high-density countries. Similarly, the increase in impervious area over the past 35 years was also concentrated in the Northern Hemisphere, and the increase between 20°N~60°N accounted for 70.75% of the total increase in the world.

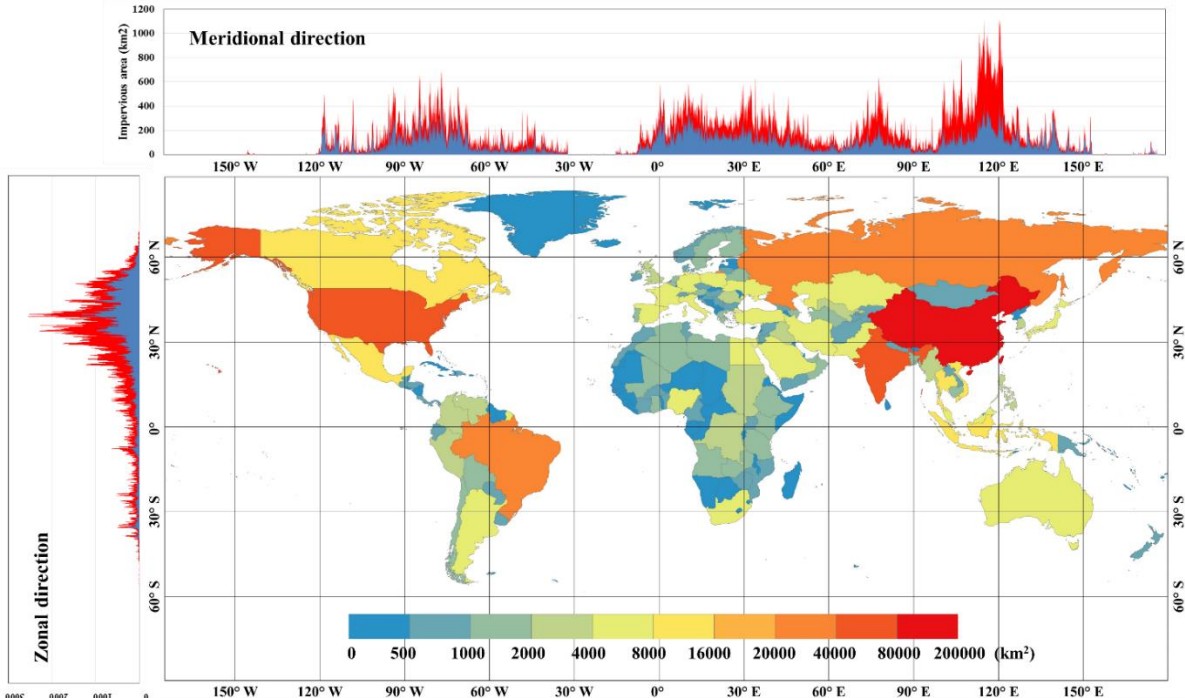

Figure 7. The expansion of impervious area in each country over the period 1985-2020, and meridional and
zonal impervious area statistics for 1985 (blue) and 2020 (red), with a step of 0.05°.

## 4.2 Accuracy assessment using validation samples

    Table 1 quantitatively assesses the performance of our time-series global impervious surface dynamic products using 23,322 multitemporal validation samples. The global impervious dynamic products achieved the overall accuracy of 90.1% and a kappa coefficient of 0.865 in the nine-strata validation system. Specifically,

from the perspective of user's accuracy, the pervious surface had the highest accuracy of 98.5% because we used the maximum impervious boundary in 2020 to monitor the impervious surface dynamics, and the prior

impervious layer in GLC_FCS30-2020 also had the high user's accuracy of 93.2% (Zhang et al., 2020). The impervious surface before 1985 achieved an accuracy of 92.3%, mainly because the stable impervious area in 1985 was obviously larger than the expanded area over each 5-year period, and capturing the expansion impervious surface was also more difficult. Furthermore, the measurements of expansion in impervious surfaces in seven 5-year periods had similar performances, with an accuracy of approximately 70%. Confusions mainly occurred in temporally adjacent periods because the transition from pervious surface to impervious surface is a slow process and spans a long period of time, which directly increases the difficulty of monitoring it. In addition, the producer's accuracy had a similar distribution law to the user's accuracy for each strata in Table 1. In addition, it can be found that the user's accuracy for the expansion of impervious surface after 2000 was higher than that before 2000, which was mainly affected by the sparser available Landsat observations before 2000 in Figure 1. Similarly, Gong et al. (2020) also found that the monitoring uncertainty before 2000 was greater than after 2000.

Table 1. The confusion matrix of our global 30 m impervious surface dynamic products using 23,322 validation samples.

| | P.S. | 1985 | 85~90 | 90~95 | 95~00 | 00~05 | 05~10 | 10~15 | 15~20 | Total | P.A. |
|---|---|---|---|---|---|---|---|---|---|---|---|
| P.S. | 9840 | 11 | 20 | 14 | 22 | 21 | 14 | 24 | 20 | 9986 | 0.985 |
| 1985 | 247 | 5408 | 61 | 49 | 41 | 17 | 20 | 8 | 5 | 5856 | 0.923 |
| 85~90 | 28 | 74 | 555 | 27 | 11 | 14 | 19 | 16 | 9 | 753 | 0.737 |
| 90~95 | 43 | 58 | 20 | 556 | 19 | 19 | 10 | 13 | 5 | 743 | 0.748 |
| 95~00 | 70 | 72 | 13 | 31 | 902 | 35 | 31 | 16 | 19 | 1189 | 0.759 |
| 00~05 | 76 | 62 | 12 | 36 | 42 | 1383 | 49 | 29 | 5 | 1694 | 0.816 |
| 05~10 | 52 | 37 | 13 | 14 | 14 | 42 | 1201 | 18 | 21 | 1412 | 0.851 |
| 10~15 | 47 | 52 | 11 | 21 | 23 | 36 | 69 | 566 | 19 | 844 | 0.671 |
| 15~20 | 55 | 59 | 8 | 7 | 14 | 21 | 30 | 43 | 608 | 845 | 0.720 |
| Total | 10268 | 5786 | 686 | 714 | 1064 | 1602 | 1435 | 662 | 689 | 23322 | |
| U.A. | 0.958 | 0.935 | 0.809 | 0.779 | 0.848 | 0.863 | 0.837 | 0.855 | 0.882 | | |
| O.A. | 0.901 | | | | | | | | | | |
| Kappa | 0.865 | | | | | | | | | | |

Note: P.S.: pervious surface; 1985: impervious surface before 1985; 85~90: expansion of impervious surface during 1985~1990; …, 15~20: expansion of impervious surface during 2015~2020; U.A.: user's accuracy; P.A.: producer's accuracy; O.A.: overall accuracy.

Figure 8 illustrates the confusion proportions of the pervious surface, the stable impervious surface and the expanded impervious surface over each 5-year period, according to the confusion matrix in Table 1. Obviously, the pervious surface and stable impervious surface before 1985 had the lowest confusion proportions, because we already knew the maximum impervious surface boundary in 2020 using multisource prior datasets. Next, the confusion proportion between the expansion of impervious surface before 2000 and the stable impervious surface in 1985 was approximately 10~20%, mainly because the Landsat imagery before 2000 was sparse, and we assumed that the land-cover information would remain stable in missing Landsat observation areas. Furthermore, there was also a certain degree of confusion between the expanded impervious surface and the pervious surface (approximately 5%), because urbanization generally occurred on the peripheries of cities, and thus was more likely to be confused with pervious surfaces. Lastly, there was also much confusion between

seven periods of impervious surface expansion, especially for the three temporally adjacent periods, because the transition from pervious surface to impervious surface is a long and slow process. Similarly, Liu et al. (2019) used the continuous change detection method to capture impervious surface dynamics and found a temporal bias between the detected change time and the actual change time.

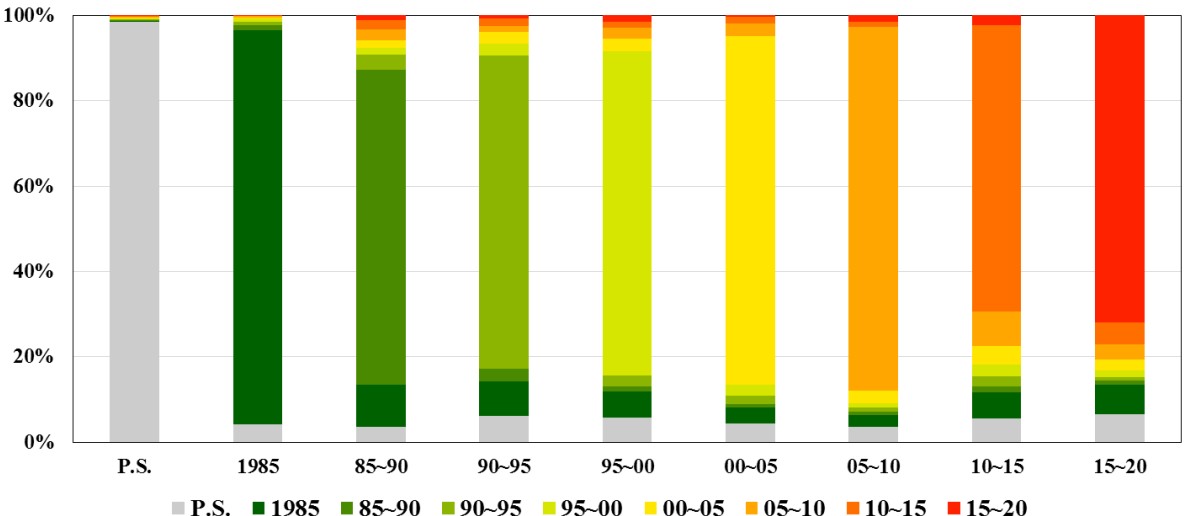

Figure 8. The confusion proportions of pervious surfaces, impervious surfaces in 1985, and increased impervious surfaces from 1985 to 2020.

## 4.3 Cross-comparisons with other global 30 m impervious surface products

### 4.3.1 Cross-comparison at global scale

To comprehensively analyze the performances of our impervious surface dynamic time-series products, five global 30 m multitemporal impervious surface products (GAIA, NUACI, GHSL, GAUD and GlobeLand30) were selected as the comparative datasets. Figure 9 illustrates the total impervious area of five global impervious surface products on six continents over the period of 1985-2020. Overall, all six global impervious surface products accurately captured the rational spatiotemporal trend over the past 35 years—the impervious surface area of all continents had steadily increased over time, and the increased impervious area in the Northern Hemisphere was obviously greater than that in the Southern Hemisphere.

Specifically, GISD30, GAIA, NUACI, GAUD and GHSL showed great area-consistency in North America, while GlobeLand30 displayed a degree of overestimation, and its estimated area was almost $0.5 \times 10^5$ km$^2$ higher than that for other products. Then, on the remaining five continents, GAIA showed the lowest total impervious area compared with the other global 30 m impervious products. Similarly, the comparison in Gong et al. (2020) also indicated that GAIA showed the lowest impervious area among several global 30 m impervious surface products (NUACI, GHSL and GlobeLand30). As the NUACI only monitored the global urban dynamics and excluded the rural areas (Liu et al., 2018), it was expected that the total impervious areas given by NUACI would be lower than those given by GISD30, GHSL and GlobeLand30. As for GHSL, its impervious area varied greatly on different continents; for example, the total impervious area was close to that of GISD30 in North America and Europe, of NUACI in Asia, South America and Oceania, and of GlobeLand30 in Africa. However, compared with the GISD30 and GlobeLand30, the GHSL still underestimated the impervious surfaces in most

continents. Next, the GlobeLand30 gave the largest total impervious area for each continent, mainly because some vegetation surfaces around buildings were regarded as artificial surfaces in GlobeLand30 (Chen et al., 2015). Lastly, the GAUD dataset showed the second lowest total impervious areas among the 6 products in Asia, South America, Africa and Oceania continents, and had the slowest impervious surface growth rates among six impervious surface products.

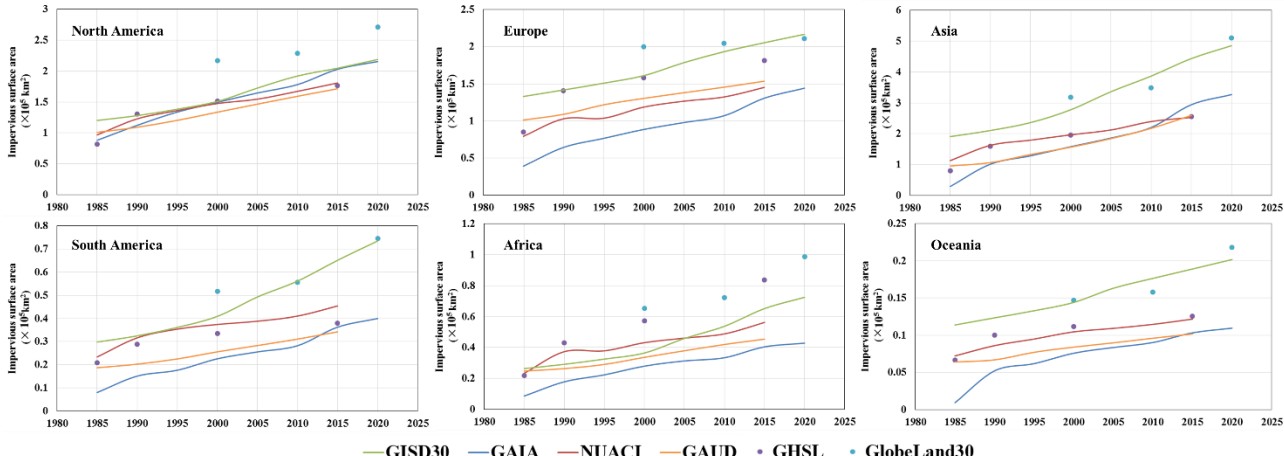

Figure 9. The impervious area of six global 30 m impervious surface products on six continents over the period of 1985-2020.

As the six global 30 m impervious surface products displayed large differences in estimated global total impervious area in Figure 9, it was necessary to further assess the performances of these products. Figure 10 illustrates the spatial patterns of these products at globe and two local enlargements in China and Europe (Figure 10a and b) after aggregating to the resolution of 0.05°. Overall, there was great spatial consistency between the GISD30, GHSL, GAUD and GlobeLand30 products—all of them captured the actual patterns of global impervious surfaces, mainly those concentrated between approximately 20°N and 60°N. Detailedly, the local enlargement in Figure 10a illustrated that GHSL showed smaller impervious areas and a lower intensity than GISD30, GAUD and GlobeLand30 in China, which meant a lot of small impervious surface pixels were underestimated by the GHSL-2015 dataset. Next, the impervious area given by GlobeLand30 in the America was greater than that given by GISD30, GAUD and GHSL, because many cities in America display a serious mix of houses and vegetation while some vegetation surfaces around buildings were regarded as artificial surfaces in GlobeLand30. It should be noted that there was highest consistency between GISD30 and GlobeLand30 in these two local enlargements. Further, the GAUD, optimized from the NUACI dataset (Liu et al., 2020b), simultaneously captured the urbans and rural areas at globe and achieved the higher performance than the NUACI dataset in two local enlargements, but it still showed lower impervious area and intensity than GISD30 and GlobeLand30 in the local regions (red rectangle regions in Figure 10a and b). Comparatively, the NUACI dataset showed the smallest impervious surface areas and the lowest intensity compared to the other products especially in Europe (Figure 10b), India and China (Figure 10a), because it only identified urban pixels and excluded rural areas (Liu et al., 2018). As for the GAIA dataset, although it simultaneously identified urban and rural pixels, their impervious surface areas were still significantly smaller than in the GISD30, GHSL, GAUD and GlobeLand30 products especially in Europe (Figure 10b), which indicated that the GAIA suffered the underestimation problem in these rural areas.

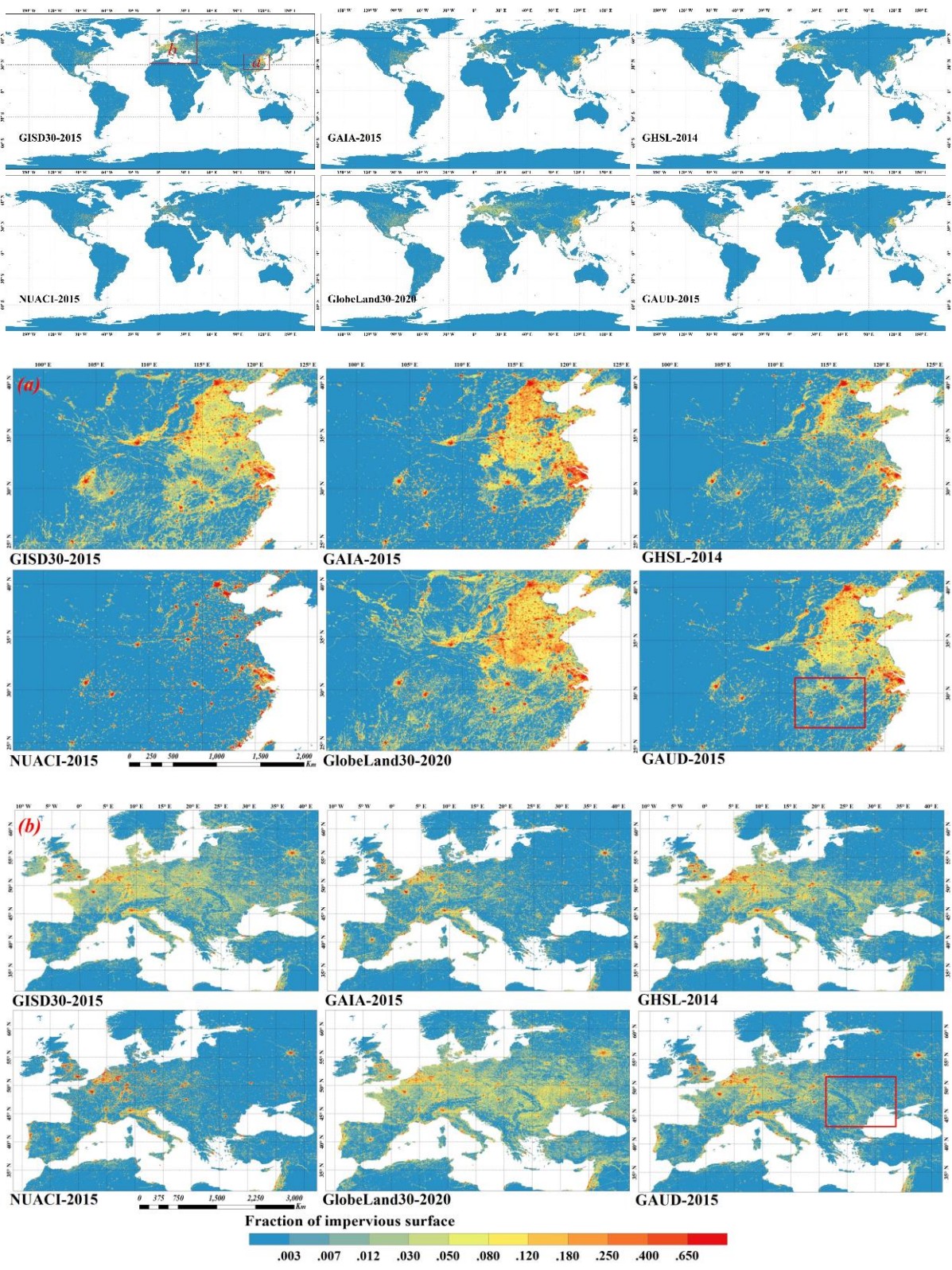

Figure 10. The spatial patterns of six global 30 m impervious surface products and two local enlargements in
China (a) and Europe (b) after aggregating to the spatial resolution of 0.05°×0.05°.

To quantitatively assess the consistency of the GISD30 dataset with five previous impervious surface products, the scatter plots and the corresponding regression functions were illustrated in the Figure 11. It should be noted that the scatter points in the Figure 11 represented the proportions of impervious area in each 0.05°×0.05° grid.

Overall, the consistency between GISD30 and other products increased with time and the regression slope also increasingly approached 1.0 (the solid regression lines were getting closer and closer to the dotted 1:1 reference line). Specifically, as for the scatter plots between GAIA and GISD30 dataset, most scatter points were obviously concentrated below the 1:1 line at early stage and then slowly distributed on both sides of the 1:1 line, and the regression slope and correlation coefficient also increased from 0.498 to 0.871 and 0.789 to 0.907, respectively. Next, as the NUACI dataset only identified the urban pixels and excluded rural areas (Liu et al., 2018), we could find that most scatter points were located below the 1:1 line especially in the 'low fraction' interval and the regression slopes were less than 1.0. Then, the scatter plots between GISD30 and GAUD datasets indicated that the impervious surfaces captured by the GISD30 was larger than that of GAUD, and the correlation coefficients and slopes between these two datasets increased with time especially in 2015 with the highest correlation coefficient of 0.931. Further, as the GlobeLand30 defined the vegetation in cities as artificial surfaces (Chen et al., 2015), we could find a lot of scatter points located above the 1:1 line. Meanwhile, as the GlobeLand30 used the minimum mapping unit of 4×4 for impervious surface (Chen et al., 2015), which meant that a large number of fragmented and small impervious surfaces were missed, the regression slopes between GlobeLand30 and GISD30 were still less than 1.0. Lastly, there was greater agreement between GISD30 and GHSL dataset than between other products in term of the spatial distributions of scatter points and the regression slope.

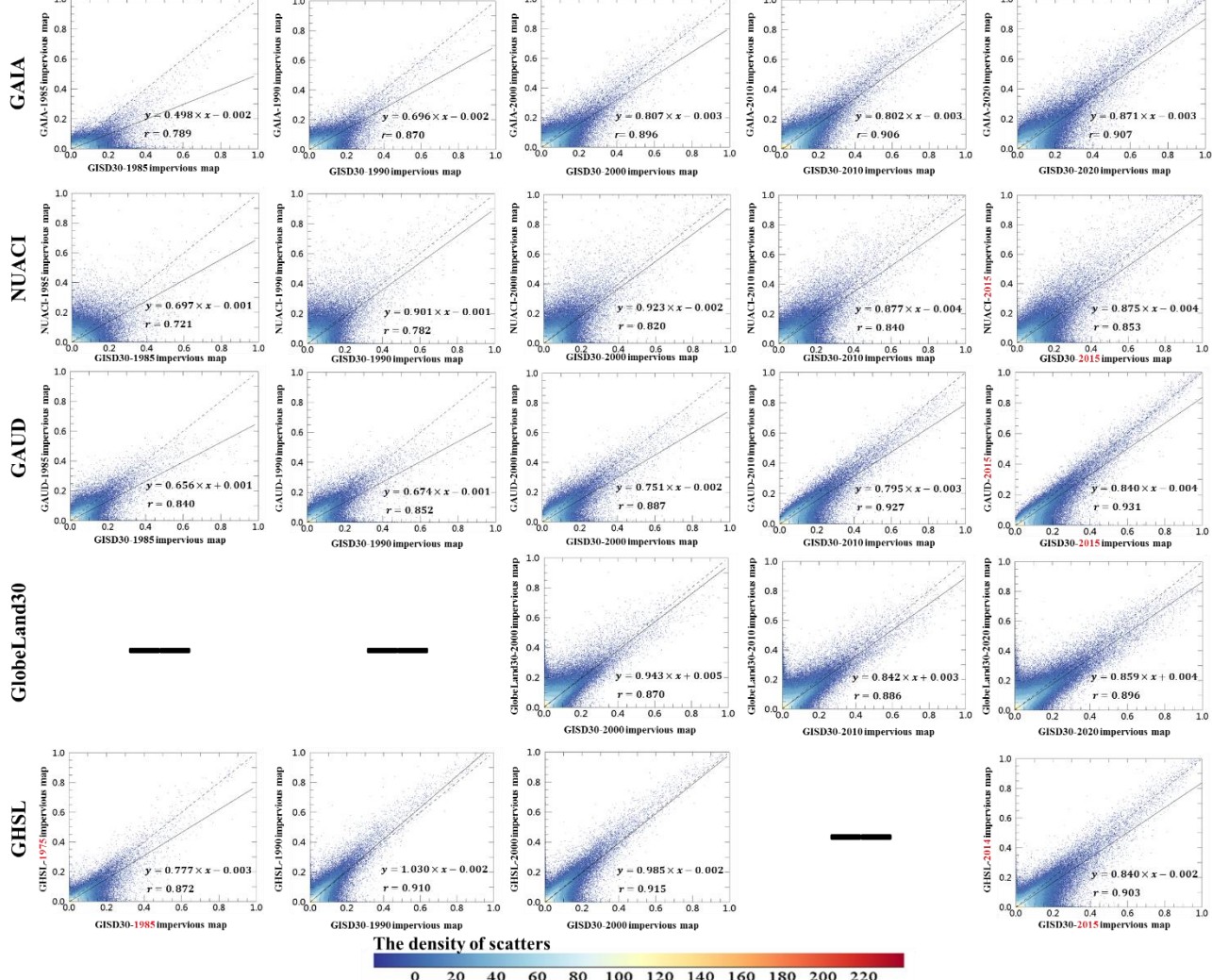

Figure 11. The scatterplots of the GISD30 dataset (x axis) against five previous global 30 m impervious surface products (y axis, GAIA, NUACI, GAUD, GlobeLand30 and GHSL datasets) at the spatial resolution of 0.05°×0.05°. It should be noted that the label of x and y axis was the proportion of impervious surfaces in each 0.05°×0.05° spatial grid.

Except for the consistency analysis, the quantitative accuracy assessments for four global impervious surface products were calculated using the same validation dataset, as listed in the Table 2. The GHSL and GlobeLand30 datasets were excluded because both of them cannot cover the whole period with 5-years interval. Overall, the GISD30 achieved the highest performance with the overall accuracy of 0.901 and kappa coefficient of 0.865, compared with 0.797 and 0.702 for GAIA, 0.843 and 0.748 for GAUD, as well as 0.745 and 0.702 for NUACI. Specifically, in terms of the pervious surfaces, it can be found that all four products achieved similar and great producer's accuracy exceeding 0.94. As the previous comparisons have illustrated that GAIA, NUACI and GAUD datasets underestimated the impervious surfaces, the user's accuracy of them was lower than the GISD30 dataset. Afterwards, as for the performances of impervious surfaces, the NUACI suffered the lowest user's accuracy and producer's accuracy in 1985 because it only identified the urban areas (Liu et al., 2018) and overestimated some increased impervious surfaces as the early impervious surfaces before 2000 (see Figure 13). Similarly, the GAIA and GAUD also missed some fragmented and small impervious surfaces, so the producer's accuracy of them in 1985 was also greatly lower than that of the GISD30. Then, the accuracy metrics of these increased impervious surfaces were similar to the overall accuracies, namely, the GISD30 could accurate capture the spatiotemporal dynamics of impervious surfaces, followed by the GAUD, GAIA and NUACI datasets.

Table 2. The accuracy metrics of four global 30 m impervious surface dynamic products using the same validation datasets.

| | | P.S. | 1985 | 85~90 | 90~95 | 95~00 | 00~05 | 05~10 | 10~15 | 15~20 | O.A. | Kappa |
|---|---|---|---|---|---|---|---|---|---|---|---|---|
| GISD30 | P.A. | 0.985 | 0.923 | 0.737 | 0.748 | 0.759 | 0.816 | 0.851 | 0.671 | 0.720 | 0.901 | 0.865 |
| | U.A. | 0.958 | 0.935 | 0.809 | 0.779 | 0.848 | 0.863 | 0.837 | 0.855 | 0.882 | | |
| GAIA | P.A. | 0.969 | 0.755 | 0.552 | 0.510 | 0.494 | 0.489 | 0.474 | 0.663 | 0.531 | 0.797 | 0.702 |
| | U.A. | 0.873 | 0.932 | 0.445 | 0.469 | 0.532 | 0.627 | 0.621 | 0.488 | 0.608 | | |
| NUACI | P.A. | 0.940 | 0.660 | 0.459 | 0.348 | 0.317 | 0.422 | 0.395 | 0.482 | | 0.745 | 0.609 |
| | U.A. | 0.839 | 0.796 | 0.160 | 0.348 | 0.398 | 0.624 | 0.626 | 0.608 | | | |
| GAUD | P.A. | 0.978 | 0.855 | 0.516 | 0.554 | 0.528 | 0.551 | 0.520 | 0.571 | | 0.843 | 0.748 |
| | U.A. | 0.896 | 0.901 | 0.535 | 0.620 | 0.642 | 0.693 | 0.637 | 0.614 | | | |

Note: P.S.: pervious surface; 1985: impervious surface before 1985; 85~90: expansion of impervious surface during 1985~1990; …, 15~20: expansion of impervious surface during 2015~2020; U.A.: user's accuracy; P.A.: producer's accuracy; O.A.: overall accuracy.

**4.3.2 Cross-comparison at regional scale**

To understand the performance of five global 30 m impervious surface products used for monitoring spatiotemporal dynamics, we randomly selected six cities after considering city size, spatial distribution and urban landscapes. Moscow and Shanghai were the representative mega cities, Bangkok and Jakarta were the cities in tropical regions (heavily affected by cloud and shadows), and Phoenix and Johannesburg were the

representative cities for arid regions. It should be noted that we excluded GlobeLand30 in regional comparisons because it only covered the period of 2000-2020 while remaining products can monitor the impervious surfaces before 2000. Specifically, Figure 12 illustrates the comparison between our GISD30 dynamic products and four comparative datasets for Moscow and Shanghai. Intuitively, NUACI suffered from overestimation for two cities, misclassifying much vegetation as the impervious surfaces. It also failed to capture the expansion of impervious surfaces in Shanghai—many cropland pixels before 2000 were identified as impervious surfaces. The GAIA products misidentified some old urban pixels (green color) as newly expanded impervious surfaces (red color) in Moscow, and it overestimated the expansion of impervious surfaces during 2010-2020 in Shanghai. According to the Landsat imagery, Shanghai's fastest urban expansion occurred in 2000-2010, but the GAIA obviously lagged in this measurement. Furthermore, GHSL also could not accurately capture the spatiotemporal dynamics of impervious surfaces in detail. For example, it gave a low proportion of expanded impervious surfaces after 2000 in Shanghai, whereas in actuality, Shanghai experienced rapid urbanization after 2000. Lastly, there was greatest spatial consistency between GAUD and GISD30 datasets in these two cities, both of them accurately captured the expansion pattern of "center-to-periphery". However, it still can be found that a lot of rural impervious surfaces in the GAUD were wrongly labeled (red rectangle) in Shanghai.

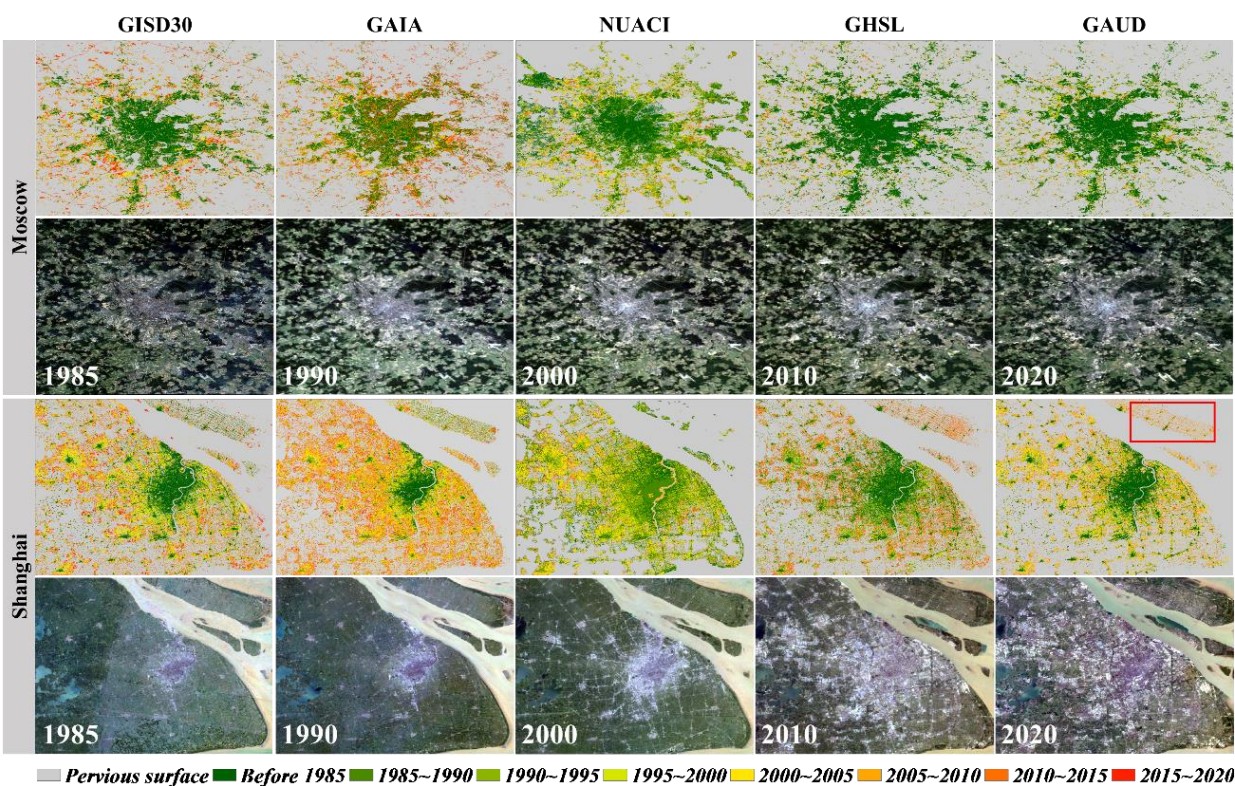

Figure 12. Comparisons between the GISD30 dynamic products and four other datasets (the GAIA products developed by Gong et al. (2020), the NUACI developed by Liu et al. (2018), the GHSL developed by Florczyk et al. (2019), and the GAUD developed by Liu et al. (2020b)) in the two representative megacities of Moscow and Shanghai. In each case, the multi-epoch Landsat imagery, comprised by red, green and blue bands, came from the United States Geological Survey (https://earthexplorer.usgs.gov/).

Figure 13 illustrates the performances of five impervious surface products in two cloud-contaminated cities (Bangkok and Jakarta). Clearly, GISD30 performed the best in monitoring the spatiotemporal dynamics of the impervious surfaces in these two cities. Comparatively, GAIA clearly underestimated the impervious surfaces

in Bangkok, and many small impervious surface objects in the peripheral cities (rural buildings) were missed. As regards impervious dynamics, GAIA underestimated the expansion after 2010 in Bangkok, and also failed to capture the expansion pattern from the city center to the outskirts in Jakarta. On the contrary, NUACI suffered from serious overestimation in two cities, and misidentified some croplands on the peripheries as impervious surfaces, especially in Jakarta. Meanwhile, it also failed to monitor the spatiotemporal dynamics of impervious surfaces in two cities, while the expansion area from 1985 to 2020 was severely underestimated and the impervious area before 2000 was overestimated. GHSL captured the distribution of impervious surfaces before 1985; however, the expansion of impervious surfaces over the past 35 years was seriously underestimated in two cities. Lastly, the GAUD dataset performed well in early stage in Bangkok, but it failed to capture the increased impervious surfaces after 2000 and missed a lot of rural impervious surfaces (red rectangle) in Bangkok. As for the second region, it still cannot accurately capture the increased impervious surfaces after 2000.

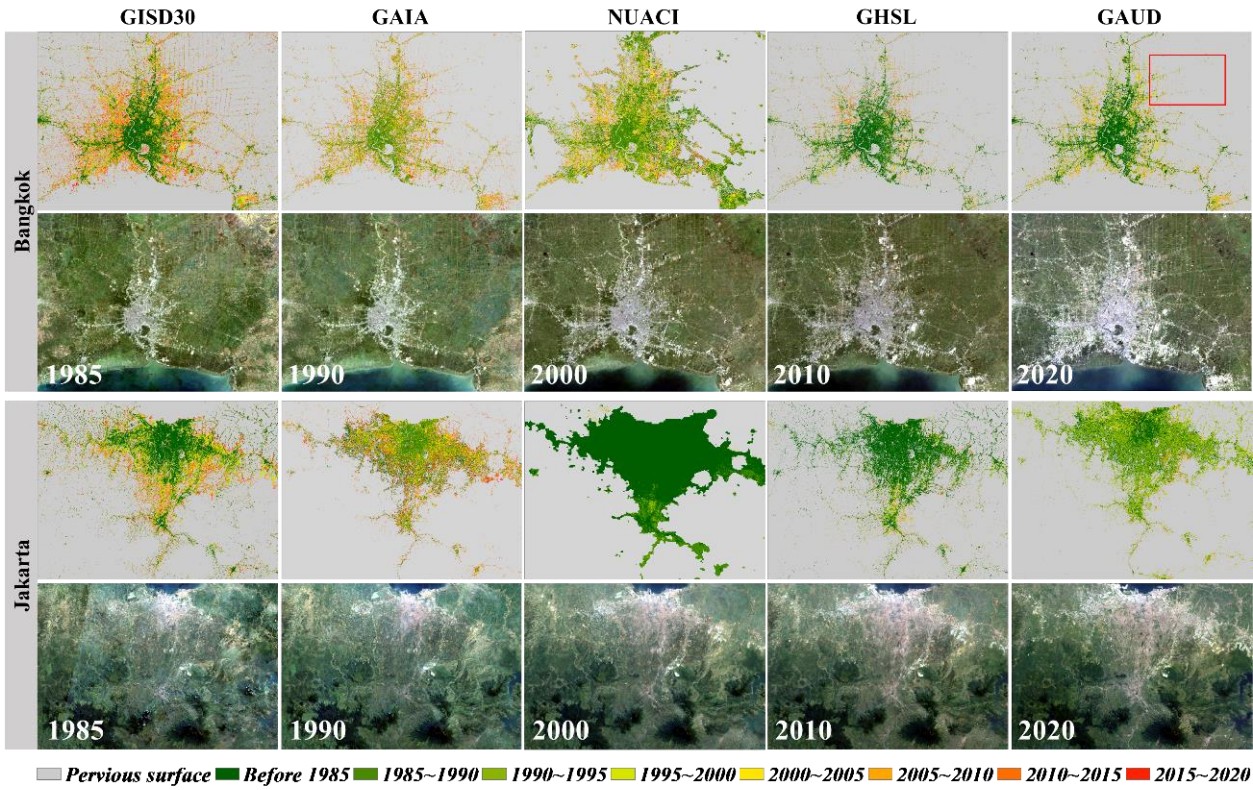

Figure 13. The comparisons between GISD30 and four reference datasets (the GAIA products developed by Gong et al. (2020), the NUACI developed by Liu et al. (2018), the GHSL developed by Florczyk et al. (2019), and the GAUD developed by Liu et al. (2020b)) in the two cloud-contaminated cities of Bangkok and Jakarta. In each case, the multi-epoch Landsat imagery, comprised by red, green and blue bands, came from the United States Geological Survey (https://earthexplorer.usgs.gov/).

Figure 14 compared the performances of our GISD30 and four reference products in two arid cities (Phoenix and Johannesburg). Overall, the highest consistency was found between GISD30 and GHSL, because both accurately captured the spatial patterns of impervious surfaces and the expansion of impervious surfaces on the peripheries of cities. NUACI showed larger impervious areas than the other four products, but the corresponding Landsat imagery indicates that NUACI misidentified many pervious surfaces (bare land) as impervious surfaces, especially in Johannesburg. Meanwhile, NUACI suffered an obvious stamping effect

mainly caused by temporal differences among adjacent Landsat image sets, and also failed to capture the time
of the expansion of impervious surfaces, especially in Johannesburg. GAIA performed well in identifying the
impervious surface area and capturing the time of expansion in Phoenix, but it suffered from overestimation in
the Johannesburg, where much arid bare land was wrongly identified as an impervious surface in the early stages.
Furthermore, as GHSL only covered the period of 1975-2014, it made sense that it registered less expanded

impervious surface than GISD30. Lastly, the GAUD shared similar impervious surface distributions with
GISD30 at early stage in Phoenix, but its increased impervious surfaces after 2000 were significantly less than
GISD30, GAIA and GHSL. As for the Johannesburg city, it suffered the overestimation problem, identifying
some pervious surfaces in the cities as the impervious surfaces, and also underestimated the increased
impervious surfaces after 2000.

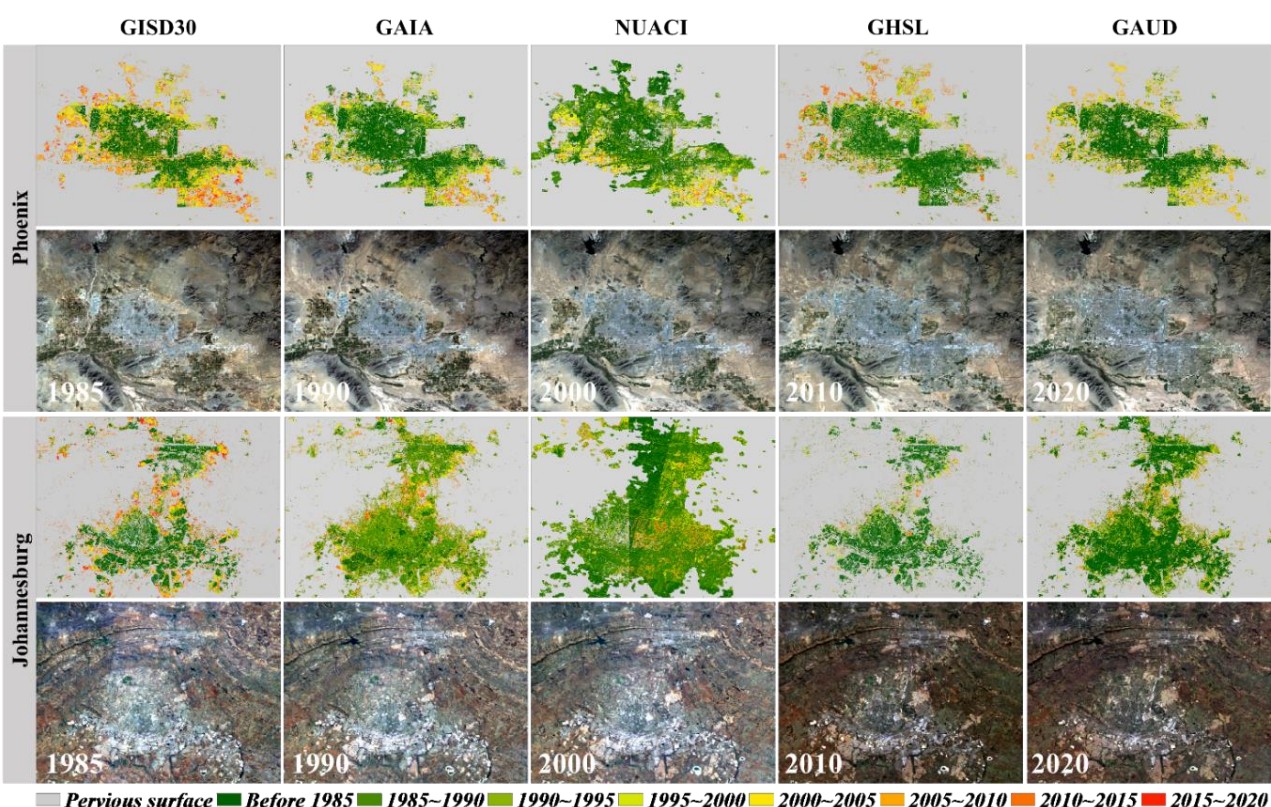

Figure 14. The comparisons between GISD30 and four reference datasets (the GAIA products developed by
Gong et al. (2020), the NUACI developed by Liu et al. (2018), the GHSL developed by Florczyk et al. (2019),
and the GAUD developed by Liu et al. (2020b)) in the two representative arid cities of Phoenix and
Johannesburg. In each case, the multi-epoch Landsat imagery, comprised by red, green and blue bands, came

from the United States Geological Survey (https://earthexplorer.usgs.gov/).

Lastly, the cross-comparison between GISD30 and four previous datasets in the rural villages (containing
sparse impervious surfaces) was illustrated in the Figure 15. Overall, except for our GISD30, the remaining
impervious surface datasets failed to identify these small rural buildings around the central villages. In terms of
the spatial pattern of villages, the NUACI dataset obviously misclassified a lot of croplands as the increased

impervious surfaces and also missed those stable impervious surfaces in the central villages. The GAUD dataset
performed well in the early stage and accurately captured these old impervious surfaces, but these increased
impervious surfaces after 2000 were missed. In fact, the village experienced significant impervious expansions
after 2000 by visually interpreting the multitemporal Landsat imagery. The GAIA partly captured the

spatiotemporal expansion in the village, but the impervious areas in the GAIA was obviously smaller than the actual situation, which indicated that the GAIA dataset suffered the underestimation problem in this rural village. Further, it can be found that there was highest consistency between GISD30 and GHSL, both of them captured the expansion pattern of "center-to-periphery", however, the increased impervious surfaces in the GHSL were still less than the actual increases.

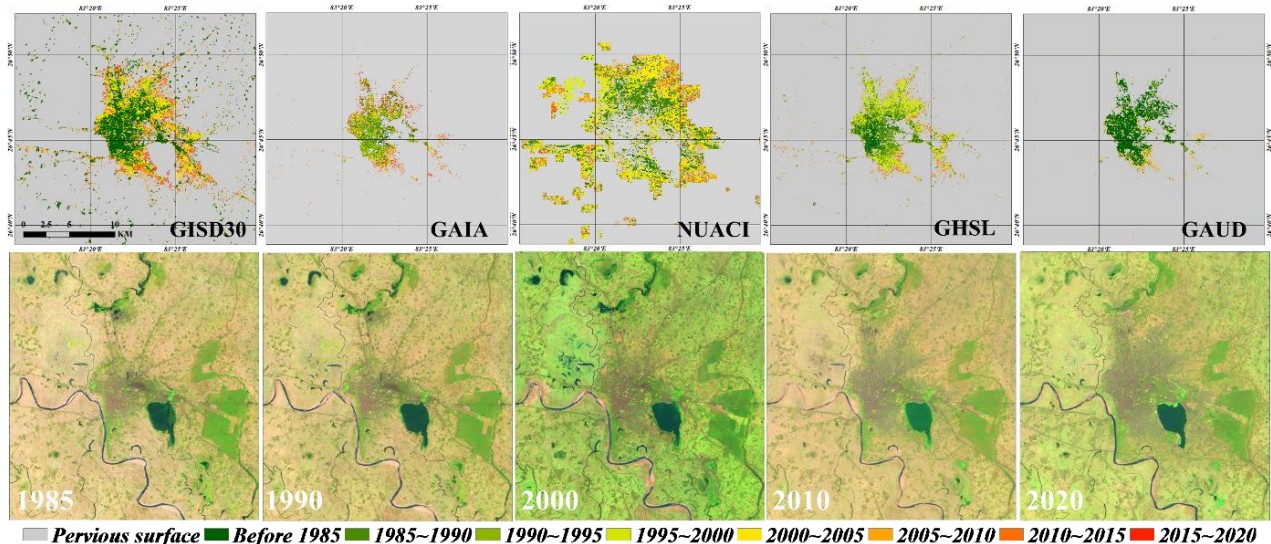

Figure 15. The comparisons between GISD30 and four reference datasets (the GAIA products developed by Gong et al. (2020), the NUACI developed by Liu et al. (2018), the GHSL developed by Florczyk et al. (2019), and the GAUD developed by Liu et al. (2020b)) in the rural village. The multi-epoch Landsat imagery, comprised by SWIR1, NIR and red bands, came from the United States Geological Survey (https://earthexplorer.usgs.gov/).

## 5 Discussion

### 5.1 The feasibility and advantages of the proposed method for monitoring impervious surface dynamics

In contrast to supervised classification methods using independent samples for different periods, which require expensive resources to collect multitemporal training samples (Gao et al., 2012; Zhang and Weng, 2016), we used prior global land-cover products and the spectral generalization strategy to automatically monitor the impervious surface dynamics. Firstly, as the reliability of the training samples was demonstrated to directly affect the final classification accuracy, we combined the impervious layers in the GLC_FCS30-2020 and GlobeLand30-2020 land-cover products to derive candidate impervious training samples, and then adopted the spatial homogeneity filtering to further ensure the reliability of each sample in 2020. In order to assess the accuracy of training samples, we randomly selected 10,000 impervious surface samples from the global sample pool, and the 10,000 random samples were interpreted by visual interpretation. The validation result showed that these impervious training samples achieved an overall accuracy of 95.52% in 2020. To demonstrate whether the erroneous training samples can affect the performance of the classifiers, we gradually increased the percentage of erroneous training samples with the step of 1 % and then repeated 100 times, illustrated in the Figure 16, it can be found that the local adaptive random forest models had great performance to be resistant to noise and erroneous training samples, and the overall accuracy and impervious surface producer's accuracy kelp stable when the percentage of erroneous training samples were controlled within 40% and then decreased after

exceeding the threshold. Similarly, Gong et al. (2019b) also found that the overall accuracy kept stable when the percentage of erroneous training samples was within 20%. Therefore, the training samples derived in Section 3.1 were accurate enough for monitoring impervious surface dynamics.

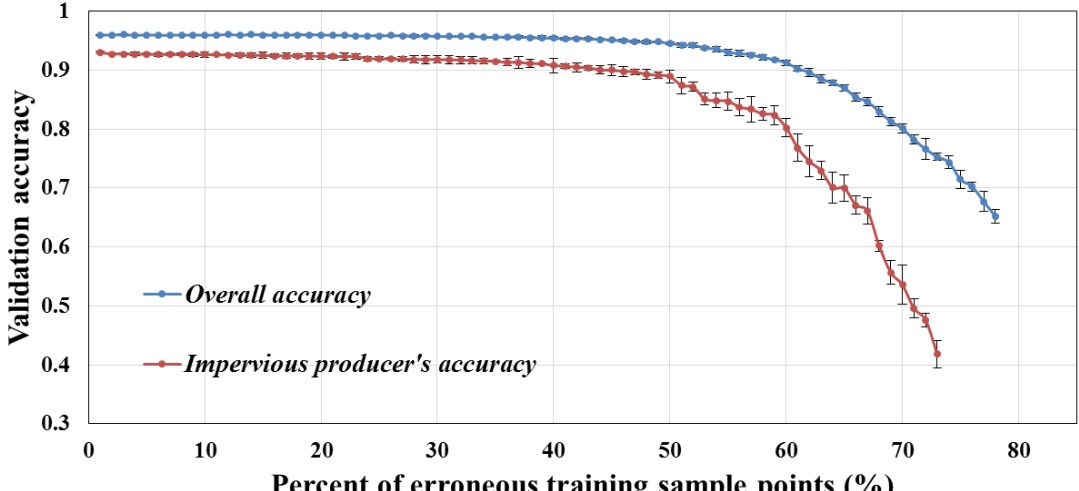

Figure 16. The relationship between overall accuracy and impervious producer's accuracy with the percentage of erroneous training samples using the random forest classification model.

In addition, contrary to other spectral generalization classification methods, which migrated the reflectance spectra of all land-cover types (Dannenberg et al., 2016; Phalke and Özdoğan, 2018; Zhang et al., 2019), we only migrated the reflectance spectra of impervious surfaces measured in 2020 to other periods, and simultaneously transferred the pervious samples to other periods based on the assumption of irreversibility. Therefore, we needed to independently train the classification models in each period using the migrated reflectance spectra of impervious and pervious surface samples. Correspondingly, our temporal adaptive models achieve better performances than traditional generalized models used for monitoring impervious surface dynamics. Furthermore, many studies have demonstrated that the spectral inconsistency between migrated spectra and classified imagery directly affects classification accuracy (Woodcock et al., 2001; Zhang et al., 2018). In this study, we used continuous Landsat imagery to preclude the effects of different sensors, and adopted a seasonally composited method with relative radiometric normalization to minimize the influence of temporal difference. We toke the Yangtze River Delta as an example to draw scatterplots for NIR reflectance of impervious surfaces in 2020 against other periods at the growing season after relative radiometric normalization illustrated in Figure 17. There were significant consistency in NIR band between reference period and other periods and most scatters were distributed on both sides of the regression line. In terms of the regression slope, the slope got closer and closer to 1.0 as time increased, which mainly caused by the shorter temporal difference and denser Landsat imagery at later periods. According to the distribution of scatter points and the regression lines, there was no systematic bias between reference data and other data, which also demonstrated that it was feasible to generalize the reflectance spectra of impervious surfaces in the 2020 to other periods.

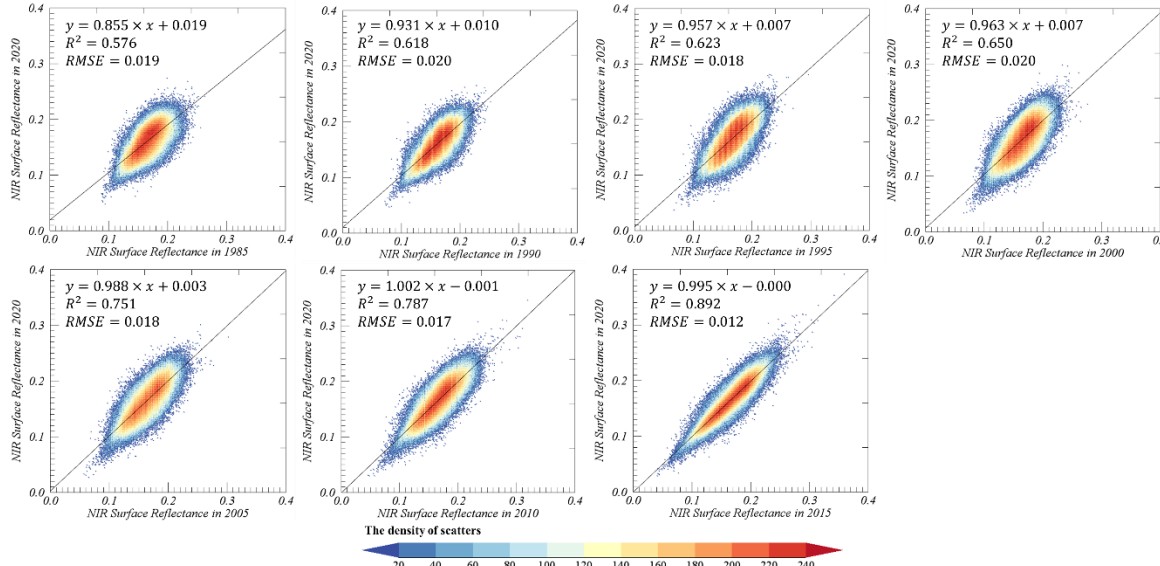

Figure 17. The scatterplots for NIR reflectance of impervious surface in 2020 (y-axis) against other periods (x-axis) after relative radiometric normalization in the Yangtze River Delta region.

Lastly, to optimize the time-series impervious maps and minimize the influence of classification error, the temporal consistency checking post-processing method proposed by the Li et al. (2015) was adopted. It mainly used the spatiotemporal correlation information to eliminate the "salt-pepper" noisy in the multi-epoch

impervious surface maps, and used the irreversible assumption to remove the illogical transitions. Li et al. (2015) quantitatively demonstrated that the post-processing method improved the overall accuracy by about 6% for monitoring impervious dynamics in Beijing, China. Recently, this post-processing method was involved for producing GAIA dataset (Gong et al., 2020) and optimizing time-series land-cover maps in China (Yang and Huang, 2021), both of them demonstrated that temporal consistency checking improved the reliability and

consistency of the classification results by integrating the spatio-temporal context information.

## 5.2 Limitations and prospects of the global impervious surface dynamic dataset

In this study, we have proposed a novel automatic method to successfully produce a global 30 m impervious surface dynamic dataset over the period of 1985-2020, and quantitatively and qualitatively demonstrated that our dataset performed well in capturing the spatial distributions and spatiotemporal dynamics of impervious

surfaces; however, there were still some weaknesses in our impervious surface dynamic products. Firstly, we assumed that the transition from pervious surface to impervious surface was irreversible over the monitoring period, which caused our method or product to fail to capture the transition from impervious to pervious surface (such as demolition caused by urban greening), as well as many changes that took place in impervious surfaces (such as urban demolition and reconstruction). Recently, Liu et al. (2019) used continuous change detection to

successfully capture these reversible and multiple changes in Nanchang, China; however, the implementation efficiency of the method was low, and whether it can support the monitoring of global impervious surface dynamics remains to be verified. Therefore, our future work must exploit the advantages of a continuous change detection model to improve the effectivity of monitoring the spatiotemporal dynamics of impervious surfaces.

Our previous study (Zhang et al., 2020) quantitatively demonstrated that a combination of multisource

remote sensing datasets could significantly improve the ability to recognize impervious surfaces, especially in

semi-arid or arid regions, where bare land generally shares spectral characteristics with impervious surfaces. In addition, the Landsat imagery available before 2000 was relatively sparse (illustrated in the Figure 1), which directly affects the monitoring accuracy of impervious surfaces, and this explains why the user's accuracy of the expansion of impervious surfaces before 2000 was significantly lower than after 2000 (Table 1). Similarly, Gong et al. (2020) also found that the availability of Landsat imagery had a positive relationship with impervious surface monitoring accuracy when creating GAIA global impervious surface products. Therefore, our future work should combine multisource remote sensing imagery (such as synthetic aperture radar (SAR), nighttime light (NTL) and AVHRR data) as auxiliary data to further improve impervious surface monitoring accuracy.

## 6    Data availability

The global 30 m impervious surface dynamic dataset from 1985 to 2020 is free to access at http://doi.org/10.5281/zenodo.5220816 (Liu et al., 2021b). The global dynamic dataset was used to label the expansion information in a single band; specifically, the pervious surface and the impervious surface before 1985 were respectively labeled 0 and l, and the expanded impervious surfaces in the periods 1985-1990, 1990-1995, 1995-2000, 2000-2005, 2005-2010, 2010-2015 and 2015-2020 were labeled 2,3,4,5,6,7 and 8. Furthermore, in order to facilitate the use of these data, the global dynamic products were split into 961 5°×5° tiles in the GeoTIFF format, named "GISD30_1985-2020_E/W**N/S**.tif", where 'E/W**N/S**' is the latitude and longitude coordinates found in the upper left corner of the tile data.

## 7    Conclusion

In this study, a novel and automatic method by combining the advantages of spectral generalization and automatic sample extraction strategy was proposed and then an accurate global 30 m impervious surface dynamic dataset for 1985 to 2020 was produced by using time-series Landsat imagery. Specifically, we first migrated the reflectance spectra of impervious surfaces, and simultaneously transferred the training samples of pervious surfaces to other periods, to automatically monitor the spatiotemporal dynamics of impervious surfaces from 1985 to 2020. Then, we combined the local adaptive modeling and time-series Landsat imagery to independently produce impervious surface time-series products. Lastly, the spatiotemporal consistency checking method was applied to independent impervious surface products in order to minimize the effects of classification errors and ensure the reliability and spatiotemporal consistency of the final impervious surface dynamic dataset.

Overall, the global 30 m impervious surface dynamic dataset we produced accurately captured the expansion pattern of impervious surfaces over the past 35 years. The quantitative results indicate that the global impervious surface area doubled in the past 35 years, from $5.116\times10^5$ km$^2$ in 1985 to $10.871\times10^5$ km$^2$ in 2020, and Asia underwent the greatest increase in impervious surface area compared to other continents, with a total increase of $2.946\times10^5$ km$^2$. Meanwhile, we also found that the expansion rate of impervious surface on six continents after 2000 was significantly faster than before 2000. In addition, the global 30 m impervious surface dynamic dataset was validated by 23,322 multitemporal validation samples, and our dataset achieved the overall accuracy of 90.1% and a kappa coefficient of 0.865. Lastly, quantitative and qualitative comparisons between our GISD30 and five comparative impervious surface products (GAIA, GHSL, NUACI, GAUD and GlobeLand30) indicate that our GISD30 products performed the best in capturing the spatial distributions and spatiotemporal dynamics of impervious surfaces. Therefore, it was concluded that our global 30 m impervious

surface dynamic dataset was an accurate product, and could provide vital support for monitoring regional or global urbanization or carrying out related tasks.

**Author contributions.** Conceptualization, Liangyun Liu; Investigation, Xiao Zhang; Methodology, Liangyun Liu and Xiao Zhang; Software, Xiao Zhang and Xidong Chen; Validation, Xiao Zhang, Tingting Zhao, Xidong Chen and Yuan Gao; Writing – original draft preparation, Xiao Zhang; writing—review and editing, Liangyun Liu.

**Competing interests.** The authors declare that they have no conflict of interest.

## Acknowledgement

We greatly appreciate the free access of GAIA product provided by the Tsinghua University, the NUACI and GAUD impervious surface products provided by Sun Yat-sen University, the GHSL impervious surface products produced by the National Aeronautics and Space Administration, and the GlobeLand30 land-cover products provided by the National Geomatics Center of China.

## Financial support

This research has been supported by the Strategic Priority Research Program of the Chinese Academy of Sciences (grant no. XDA19090125), the Key Research Program of the Chinese Academy of Sciences (grant no. ZDRW-ZS-2019-1), and the National Natural Science Foundation of China (grant no. 41825002).

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
