# Peer review of "GISD30: global 30-m impervious surface dynamic dataset from 1985 to 2020 using time-series Landsat imagery on the Google Earth Engine platform"

_Earth System Science Data, 2021_

## Author Comment (AC1)

**Response to comments**

**Paper #:** essd-2021-285
**Title:** GISD30: global 30-m impervious surface dynamic dataset from 1985 to 2020 using time-series Landsat imagery on the Google Earth Engine platform
**Journal**: Earth System Science Data

**Reviewer #1**

This manuscript was trying to derive a new time-series (every five-year interval from 1985 to 2020) impervious surface dataset from Landsat imagery with the aid of the Google Earth Engine (GEE) platform. The authors divided the land surface into 961 5°×5° geographical tiles, and used random forest classifiers to identify impervious surfaces in each tile and period. Then they adopt a temporal consistency correction method to smooth the independent results classified during different five-year periods. A satisfactory performance was claimed (an overall accuracy of 91.5% and a kappa coefficient of 0.866) using more than 18 thousand validation samples. However, there are a number of concerns on the framing and introduction, the scientific contribution to literature and current datasets, the clarity of the methodology and results, and most importantly, the validation of the derived dataset. Below I highlight these key areas.

Great thanks for the detailed and useful comments. The manuscript has been greatly improved according to your and another reviewer's comments.

**The frame of the introduction**

In the introduction, the authors reviewed many previous studies and existing impervious surface products, and summarized a series of existing problems, such as significant inconsistency and uncertainty within existing datasets (L65-69), the monitoring efficiency of the time-series change detection strategy being very low (L81-82, not necessarily true), image classification strategy performing well but collecting training samples being time-consuming and labor-intensive (L94-95). However, all these problems raised are not solved in the current version of manuscript and the objective of this study is missing. Are the authors going to reduce the uncertainty in inconsistent areas, or trying to improve the efficiency of the classification procedure?

Great thanks for pointing out the issue. Yes, we raised a lot of problems after reviewing many previous studies and existing impervious surface products. In this study, we proposed a novel and automatic method to improve the efficiency of the classification procedure and then produced an accurate global 30 m impervious surface dynamic datasets.

Firstly, the problem of significant inconsistency and uncertainty within existing datasets in the Introduction section has been removed and revised as:

[revised manuscript text omitted]

There have been already serval impervious surface products available, and why the authors still tried to derived a new one?

Great thanks for pointing out the issue. The reasons why we derive a new impervious dynamic dataset has been revised as:

"many global 30 m multitemporal impervious surface products have been produced using long time-series Landsat imagery (Florczyk et al., 2019a; Gong et al., 2020; Liu et al., 2018; Liu et al., 2020b). Liu et al. (2021a) comprehensively reviewed current seven global 30 m impervious surface products, and found **only four products could capture the impervious expansion at the long time-series**. Specifically, Liu et al. (2018) proposed a new index to develop multitemporal global 30 m urban land maps for 1990 to 2010 with 5-years intervals, **but the products suffered the low producer's accuracy and user's accuracy of 0.50–0.60 and 0.49–0.61**. Gong et al. (2020) used a combination of "exclusion–inclusion" and "temporal check" methods to generate the first annual global 30 m artificial impervious surface area dataset for 1985 to 2018, **but the cross-comparisons in the Zhang et al. (2020) found that this annual dataset achieved great performance on mega-cities but suffered the under-estimation problems in the rural areas**. T**he global human settlement layer (GHSL) monitored the impervious dynamic from 1975 to 2015 (Florczyk et al., 2019b), but it suffered the overestimation problems at early stage and also missed the fragmented impervious objects (Gong et al., 2020)**. Therefore, **an accurate global 30 m impervious surface dynamic product, which could accurately capture the spatiotemporal dynamic of various impervious objects including cities and rural**, is still urgently needed."

The text in the last paragraph of the introduction "The aim of the study was to automatically produce an accurate and novel global 30 m impervious surface dynamic dataset (GISD30) for 1985 to 2020…" (L102-104) is more like a task rather than a study to solve particular scientific problem(s). I suggest that in the introduction the authors need to focus more on the what is/are the most urgent problem(s) in current studies, what are the main challenges behind these problems, and why the proposed methods / strategies are capable of solving these problems.

Great thanks for the suggestion. Based on the suggestion, the last paragraph has been revised as:

"Monitoring impervious surface dynamics is a challenging and time-consuming task due to its high spatiotemporal heterogeneity. In this study, we proposed to a novel and automatic method by combining the

advantages of spectral generalization and automatic sample extraction strategy for monitoring time-series impervious surface dynamics. Specifically, we derived the training samples from prior land-cover products to solve the time-consuming and manual participation problems for manually collecting massive training samples. Then, we combined the derived training samples with the temporally spectral generalization to independently mapping impervious surfaces at long time-series. Next, a spatiotemporal consistency correction method was applied to independent impervious surface maps to minimize the effects of classification errors and ensure the reliability and spatiotemporal consistency of the final dynamic impervious surface dataset. Finally, we produced an accurate and novel global 30 m impervious surface dynamic dataset (GISD30) from 1985 to 2020 by combining the proposed method and Google Earth Engine cloud computing platform, which also provide vital support for monitoring regional or global urbanization and performing related tasks."

**The methodology**

1. As described in L135, the authors extracted the impervious surfaces in 2020 from the GLC_FCS30-2020 (Zhang, et al., 2021) and used it as a baseline to derive the time-series impervious surfaces from 1985 to 2020. It is not clear what is the difference between this extracted impervious surface layer and the impervious surfaces derived by the authors. Is the extracted impervious surface layer directly used as the result for period of 2015-2020?

Great thanks for the comment. It is different between the derived impervious surface maps during 2015-2020 and the prior impervious surface layer in GLC_FCS30-2020. Specifically, the impervious layer in GLC_FCS30-2020 was only one of dataset to derive impervious surface training samples and the maximum impervious surface extents, while the derived impervious surface map during 2015-2020 was developed by the local adaptive classification models and then optimized by the temporal consistency checking algorithm. The explanation has been added in the revised manuscript as:

It should be noted that the impervious surface layer in the GLC_FCS30-2020 dataset, which was independently produced by combining multisource and multitemporal remote sensing imagery and achieved an overall accuracy of 95.1% and a kappa coefficient of 0.898 (Zhang et al., 2020), was not used as the result for period of 2015-2020 in the final results, instead, only used as the prior dataset for deriving training samples and determining the broadest extents.

2. The authors mentioned plenty times of their previous studies in the methodology section, such as Zhang et al., 2018, Zhang et al., 2019, Zhang et al., 2020, and Zhang et al., 2021. I am aware that the method used in this manuscript was developed based upon their previous ones. But most of these descriptions should be moved to the introduction and leave only the most original ones in the methodology section. Again, many discussions on other previous studies are found in the methodology section, too. They should also be moved to the introduction. In the current version, I can hardly see the core method proposed by the author or the adaptations/modifications of previous methods to derive time-series impervious surface.

Great thanks for the comment. According to the suggestion, the previous works have been moved to the introduction. The method section has been totally improved and rewritten as:

[revised manuscript text omitted]

3. Perhaps I missed it but I couldn't see how many training samples the authors used to calibrate the random forest classifier in each 5°×5° tile and five-year interval.

Thanks for the comment. The sample size in each geographical tile was missed in the previous manuscript. Actually, the sample size in each tile was approximately 20,000 when the local region simultaneously contained the impervious surfaces, bare land, cropland and other pervious land-cover types. In the revised manuscript, we added the sample size in each tile as:

"As for the sample size in each tile, [Zhu et al. (2016)]() quantitatively demonstrated that the mapping accuracy first increased and then stabilized with the increase of the sample size and suggested a minimum of 600 training samples and a maximum of 8000 training samples per class. In this study, the sample size is about 5000 for each class, and the ratio between impervious surfaces and pervious surfaces is 1:3."

Meanwhile, we only derived training samples one time in 2020 excluding other periods, because we transferred the pervious samples in 2020 to other periods using the land-cover irreversible assumption, then, we proposed to temporally generalize the reflectance spectra of impervious surfaces in 2020 to other periods after relative radiometric normalization to solve the lack of samples for impervious surfaces before 2020. We combined the temporally spectral generalization and deriving training samples from existing products to achieve the automatically monitoring impervious surface dynamics. It has been added in the Section 3.2 as:

"Based on the assumption that the land-cover transition from impervious surface to pervious surface was irreversible, the derived pervious samples in 2020 (Section 3.1.1) would be directly transferred to other periods, but the impervious surface samples in 2020 cannot be transferred. To solve the lack of samples for impervious surfaces before 2020, we normalized the reflectance spectra of impervious surfaces in other epochs to those in 2020 using the relative radiometric normalization method (Section 3.1.2)"

4. According to the manuscript, the global training samples were automatically collected from the GLC_FCS30-2020 land-cover product. Did the authors check the reliability of the labels in these training samples? Even though a high accuracy was achieved in this GLC_FCS30-2020 product, false labels still exist when the training samples are located in incorrect detected areas (impervious surfaces, cropland, bare land, and other pervious surfaces). These training samples with false label can directly bias the classifiers which were later on used to identify the time-series impervious surfaces.

Great thanks for the comment. Yes, we have checked the reliability of the confidence of these training samples, and found that these derived impervious training samples in 2020 achieved the overall accuracy of 95.52% in the Section 5.1. As for the erroneous training samples, we have added the qualitative analysis of the relationship between overall accuracy and impervious producer's accuracy with the percentage of erroneous training samples in the Section 5.1 as:

In order to assess the accuracy of all training samples, we randomly selected 10,000 impervious surface samples from the global sample pool, and the 10,000 random samples were interpreted by visual interpretation. The validation result showed that these impervious training samples achieved an overall accuracy of 95.52% in 2020. To demonstrate whether the erroneous training samples can affect the performance of the classifiers, we gradually increased the percentage of erroneous training samples with the step of 1 % and then repeated 100 times in Figure 16, it can be found that the local adaptive random forest models had great performance to be resistant to noise and erroneous training samples, and the overall accuracy and impervious surface producer's accuracy kelp stable when the percentage of erroneous training samples were controlled within 40% and then obviously decreased after exceeding the threshold. Similarly, Gong et al. (2019b) also found that the overall accuracy kept stable when the percentage of erroneous training samples was within 20%. Therefore, the training samples derived in Section 3.1 were accurate enough for monitoring impervious surface dynamics.

[Figure]

Figure 16. The relationship between overall accuracy and impervious producer's accuracy with the percentage of erroneous training samples using the random forest classification model.

5. Besides, the equation in L264 is unnumbered, and it is the only equation throughout the manuscript. I believe this is not rigorous enough for a manuscript considering for publication in ESSD. I suggest the authors add more equations to better describe the techniques used in this study.

Great thanks for the comment. The formula number has been added. Based on your suggestion, the method section has been totally improved and rewritten and more equations has been added in the revised manuscript as:

**Section 3.1.1**

[revised manuscript text omitted]
 (L311-312). How did they collect the training samples and obtain the labels in period other than 2020?
Great thanks for the comment. The novelty of our method was to derive training samples one time in 2020 by combine temporally spectral generalization and deriving training samples from existing products for monitoring impervious surface dynamics. Specifically, we transferred the pervious samples in 2020 to other periods using the land-cover irreversible assumption, then temporally generalized the reflectance spectra of impervious surfaces in 2020 to other periods after relative radiometric normalization to solve the lack of impervious surfaces samples before 2020. The explanation has been added in the revised manuscript as:
Based on the assumption that the land-cover transition from impervious surface to pervious surface was irreversible, the derived pervious samples in 2020 (Section 3.1.1) would be directly transferred to other periods, but the impervious surface samples in 2020 cannot be transferred. To solve the lack of impervious surface samples before 2020, we normalized the reflectance spectra of impervious surfaces in other epochs to those in 2020 using the relative radiometric normalization method (Section 3.1.2). Specifically, we independently trained the classification models at each period using the generalized impervious reflectance spectra ($TrainFeatures\_IS_{2020}$) and the pervious samples ($TrainFeatures\_PS_t$) as:

$$TrainFeatures\_PS_t = \left[\sum_{s_i}\left(\rho_b^{s_i,t}, \rho_g^{s_i,t}, \rho_r^{s_i,t}, \rho_{nir}^{s_i,t}, \rho_{swir1}^{s_i,t}, \rho_{swir2}^{s_i,t}, ndbi^{s_i,t}, ndvi^{s_i,t}, ndwi^{s_i,t}\right)\right] \qquad (5)$$

$$TrainFeatures\_IS_{2020} = \left[\sum_{s_i}\left(\rho_b^{s_i}, \rho_g^{s_i}, \rho_r^{s_i}, \rho_{nir}^{s_i}, \rho_{swir1}^{s_i}, \rho_{swir2}^{s_i}, ndbi^{s_i}, ndvi^{s_i}, ndwi^{s_i}\right)\right]$$

where $s_i$ denotes various seasonal composites and $t$ is the monitored period. It can be found that the $TrainFeatures\_PS_t$ varies with the $t$, namely, the training spectra of pervious surfaces came from the unclassified imagery. It should be noted that there may not be cloud-free imagery available especially for the rainy season before 2000 in the tropical rainforest areas. In this case, we discarded this missed seasonal features when training the classification models, namely, the number of training features varied with the availability of Landsat observations.

7. How to deal with the situation where there is no Landsat imagery available? The author presented the availability of Landsat observation during each five-year interval (figure 1).

Great thanks for the comment. Firstly, as for these no Landsat observation areas, we assumed the land-cover in these areas would remain stable. According to our statistics in Figure 1, we found the missing Landsat observations during 1986 to 1995 mainly concentrated on the Northeast Asia where contained a small number of impervious surfaces, so the unchanged assumption had little effect in these time periods. Similarly, almost all existing time-series land-cover products (including impervious surface products, GAUD and GAIA) also used the assumption in those no observation areas. The assumption has been added in the Section 3.1.2 as:

It should be noted that we categorized the time-series Landsat imagery from 1982 to 2020 into 8 periods with the interval of 5 years corresponding to the Figure 1. Meanwhile, we also assumed that the land-cover in those no Landsat observation areas would remain stable during the period. Fortunately, the missing Landsat observations during 1986 to 1995 mainly concentrated on the Northeast Asia where contained a small number of impervious surfaces, so the unchanged assumption had little effect in these time periods.

However, the proposed method uses seasonally composited imageries as input features of the classifier (L243-247), and there may not be cloud-free imagery available especially for the rainy season before 2000 in the tropical rainforest areas.

Great thanks for the comment. Yes, there may not be cloud-free imagery available especially for the rainy season before 2000 in the tropical rainforest areas. In this study, we discarded the corresponding seasonal composites if there were no enough Landsat observations to derive the seasonal composites. Namely, we might use two or three seasonal composites (except for the rainy season) as the input features to train the classifiers. The statement has been added in the revised manuscript as:

Based on the assumption that the land-cover transition from impervious surface to pervious surface was irreversible, the derived pervious samples in 2020 (Section 3.1.1) would be directly transferred to other periods, but the impervious surface samples in 2020 cannot be transferred. To solve the lack of impervious surface samples before 2020, we normalized the reflectance spectra of impervious surfaces in other epochs to those in 2020 using the relative radiometric normalization method (Section 3.1.2). Specifically, we independently trained the classification models at each period using the generalized impervious reflectance spectra ($TrainFeatures\_IS_{2020}$) and the pervious samples ($TrainFeatures\_PS_t$) as:

$$TrainFeatures\_PS_t = \left[\sum_{s_i}\left(\rho_b^{s_i,t}, \rho_g^{s_i,t}, \rho_r^{s_i,t}, \rho_{nir}^{s_i,t}, \rho_{swir1}^{s_i,t}, \rho_{swir2}^{s_i,t}, ndbi^{s_i,t}, ndvi^{s_i,t}, ndwi^{s_i,t}\right)\right]$$

$$TrainFeatures\_IS_{2020} = \left[\sum_{s_i}\left(\rho_b^{s_i}, \rho_g^{s_i}, \rho_r^{s_i}, \rho_{nir}^{s_i}, \rho_{swir1}^{s_i}, \rho_{swir2}^{s_i}, ndbi^{s_i}, ndvi^{s_i}, ndwi^{s_i}\right)\right]$$

(5)

where $s_i$ denotes various seasonal composites and $t$ is the monitored period. It can be found that the $TrainFeatures\_PS_t$ varies with the $t$, namely, the training spectra of pervious surfaces came from the

unclassified imagery. **It should be noted that there may not be cloud-free imagery available especially for the rainy season before 2000 in the tropical rainforest areas. In this case, we discarded this missed seasonal features when training the classification models, namely, the number of training features varied with the availability of Landsat observations.**

8. The authors did not mention the exact time-span of Landsat imagery used to derive each impervious surface layer. I guess the time-spans correspond to the periods presented in figure 1. For example, they used Landsat imageries during 1991-1995 to derive the impervious surface layer for 1995. But I believe it is more reasonable to use imageries before and after the target year, for example, using imageries during 1993-1997 to derive the impervious surface layer for 1995.

Great thanks for the comment. The time-span of Landsat imagery used to derive each impervious surface layer has been added in the revised manuscript as:

It should be noted that we categorized the time-series Landsat imagery from 1982 to 2020 into 8 periods with the interval of 5 years corresponding to the Figure 1. Meanwhile, we also assumed that the land-cover in those no Landsat observation areas would remain stable during the period. Fortunately, the missing Landsat observations during 1986 to 1995 mainly concentrated on the Northeast Asia where contained a small number of impervious surfaces, so the unchanged assumption had little effect in these time periods.

Further, we greatly appreciate the suggestion of using the imagery before and after the target year for mapping impervious surfaces in target year. Our next updated version of the products would adopt the suggestion.

**The results**

1. The spatial distribution of the time-series global impervious surface presented in figure 5 does not provide much spatial details nor temporal dynamics for the impervious surface because of relatively small size of impervious surface expansion compared to the entire terrestrial surface. Local enlargements of hotspot area should be presented here for better illustration of the results.

Great thanks for the suggestion. The local enlargements of hotspot areas in China and India have been added in the Figure 5 and the explanation about the figure has also revised as:

"Figure 5 illustrates the spatial distributions of time-series global 30 m impervious surface maps and two local enlargements in China and India during 1985-2020 with intervals of 5 years. Intuitively, as the world's main impervious surfaces and economic activities are mainly concentrated in the northern hemisphere, the intensity of impervious surface expansion in the northern hemisphere is more significant than that in the southern hemisphere. Specifically, the impervious surfaces have undergone rapid urbanization in past 35 years especially in developing countries such as China and India in Figure 5a and b. It can be found that many low-density areas in 1985 were transformed into medium/high-density areas in 2020, the cities were obviously connected by the new impervious surfaces especially in the mega-cities such as Shanghai and Guangzhou in China, and mega-cities (Bangkok, New Delhi and Beijing in Figure 5a and b) experienced faster impervious surface expansion than the surrounding villages, small cities, etc."

[Figure]

Figure 5. The spatial distributions of time-series global 30 m impervious surface results and two local enlargements from 1985 to 2020 with intervals of 5 years. Each pixel represents the fraction of impervious surface within each 0.05°×0.05° spatial grid.

2. Again, the comparison between the derived pattern and other available products in figure 10 is not clear from the global view. Local enlargements should be presented, too.

Great thanks for the suggestion. Two local enlargements in China and Europe have been added and the explanation about the Figure has been revised as:

As the six global 30 m impervious surface products displayed large differences in estimated global total impervious area in Figure 9, it was necessary to further assess the performances of these products. Figure 10 illustrates the spatial patterns of these products at globe and two local enlargements in China and Europe (Figure 10a and b) after aggregating to the resolution of 0.05°. Overall, there was great spatial consistency between the GISD30, GHSL, GAUD and GlobeLand30 products—all of them captured the actual patterns of global

impervious surfaces, mainly those concentrated between approximately 20°N and 60°N. Detailedly, the local enlargement in Figure 10a illustrated that GHSL showed smaller impervious areas and a lower intensity than GISD30, GAUD and GlobeLand30 in China, which meant a lot of small impervious surface pixels were underestimated by the GHSL-2015 dataset. Next, the impervious area given by GlobeLand30 in the America was greater than that given by GISD30, GAUD and GHSL, because many cities in America display a serious mix of houses and vegetation while some vegetation surfaces around buildings were regarded as artificial surfaces in GlobeLand30. It should be noted that there was highest consistency between GISD30 and GlobeLand30 in these two local enlargements. Further, the GAUD, optimized from the NUACI dataset (Liu et al., 2020b), simultaneously captured the urbans and rural areas at globe and achieved the higher performance than the NUACI dataset in two local enlargements, but it still showed lower impervious area and intensity than GISD30 and GlobeLand30 in the local regions (red rectangle regions in Figure 10a and b). Comparatively, the NUACI dataset showed the smallest impervious surface areas and the lowest intensity compared to the other products especially in Europe (Figure 10b), India and China (Figure 10a), because it only identified urban pixels and excluded rural areas (Liu et al., 2018). As for the GAIA dataset, although it simultaneously identified urban and rural pixels, their impervious surface areas were still significantly smaller than in the GISD30, GHSL, GAUD and GlobeLand30 products especially in Europe (Figure 10b), which indicated that the GAIA suffered the underestimation problem in these rural areas.

[Figure]

[Figure]

Figure 10. The spatial patterns of six global 30 m impervious surface products and two local enlargements in China (a) and Europe (b) after aggregating to the resolution of 0.5°×0.5°.

3. There are several problems/confusions in the scatter plots in figure 11. The authors did not give clear labels to the axes. I guess they refers to the proportions of impervious surface after aggregation into the coarse resolution. The color map below the scatter plots is not clear, too. Maybe the color refers to the scatter density in the plots. The root-mean-square error (RMSE) and the coefficient of determination (R2) presented here are mathematically/statistically incorrect, because these two indicators are usually used as a measure of how well the reference data (observed outcomes) are replicated by the model outputs, while the compared products here are not ground truths. Instead, the authors should present the measure of correlation coefficients.

Great thanks for the suggestion. Yes, the label of x and y axis was the proportions of impervious surface in each 0.05°×0.05° spatial grid, it has been added in the title of the Figure 11. Then, the color referred to the scatter density in the plots, it has added in the Figure 11. And two metrics have been replaced by the correlation coefficients based on the comment as:

[Figure]

Figure 11. The scatter plots between our GISD30 dataset (x axis) and five global 30 m impervious surface products (y axis, GAIA, NUACI, GAUD, GlobeLand30 and GHSL) at the spatial resolution of 0.05°×0.05°. It should be noted that the label of x and y axis was the proportion of impervious surface in each 0.05°×0.05° spatial grid.

I don't know the exact geographical distributions of the scatters, but I believe the numerical distribution of the scatters is problematic. There are too many scatters located at/closed to 0-value, while only a small proportion of them fall within the range of 20%-100%. This will largely bias the slope and intercept of the fitting lines. The authors should reduce the number of 0-value scatters and at the same time increase the number of scatters with higher values (do not include too scatters with 100% value).

Great thanks for the comment. After carefully checking, there was no error in the numerical distribution of the scatters. Firstly, we derived the scatters from the fraction of impervious surface maps at the spatial resolution of 0.05°×0.05°, and used the rule of whether the impervious pixel in each dataset was greater than 0. Then, the reason why there were too many scatters closed to 0-value was because broken country houses and small villages were more widely distributed than large cities over the globe. As for the suggestion of reducing the number of 0-value scatters, as Figure 11 was used to analyze the consistency of the GISD30 dataset with five previous impervious surface products at the global scale, we cannot artificially adjust the distribution of scatter points. To make the Figure 11 clearer, the explanation of the Figure 11 has been revised as:

To quantitatively assess the consistency of the GISD30 dataset with five previous impervious surface products, the scatter plots and the corresponding regression functions were illustrated in the Figure 11. It should be noted that the scatter points in the Figure 11 represented the proportions of impervious area in each 0.05°×0.05° grid. Overall, the consistency between GISD30 and other products increased with time, and the regression slope also increasingly approached 1.0 (the solid regression lines were getting closer and closer to the dotted 1:1 reference line). Specifically, as for the scatter plots between GAIA and GISD30 dataset, most scatter points were obviously concentrated below the 1:1 line at early stage and then slowly distributed on both sides of the 1:1 line, and the regression slope and correlation coefficient also increased from 0.498 to 0.871 and 0.789 to 0.907, respectively. Next, as the NUACI dataset only identified the urban pixels and excluded rural areas (Liu et al., 2018), we could find that most scatter points were located below the 1:1 line especially in the 'low fraction' interval and the regression slopes were less than 1.0. Then, the scatter plots between GISD30 and GAUD datasets indicated that the impervious surfaces captured by the GISD30 was larger than that of GAUD, and the correlation coefficients and slopes between these two datasets increased with time especially in 2015 with the highest correlation coefficient of 0.931. Further, as the GlobeLand30 defined the vegetation in cities as artificial surfaces (Chen et al., 2015), we could find a lot of scatter points located above the 1:1 line. Meanwhile, as the GlobeLand30 used the minimum mapping unit of 4×4 for impervious surface (Chen et al., 2015), which meant that a large number of fragmented and small impervious surfaces were missed, the regression slopes between GlobeLand30 and GISD30 were still less than 1.0. Lastly, there was greater agreement between GISD30 and GHSL dataset than between other products in term of the spatial distributions of scatter points and the regression slope.

4. The paragraph related to figure 11 (L509-520) should be rephrased to explain why the derived results yields smaller proportions of impervious surface compared with all other pervious products. Are the derived results underestimate the actual situation? Moreover, it confuses me that the results in figure 11 are somewhat contradictory to the results in figure 9. According to results in figure 9, the areas of the derived impervious surface are larger than those of the GAIA, NUACI, and GHSL across different continents. But the results in figure 11 show smaller proportions of derived impervious surface.

Great thanks for the comment. I think the comment is a bit misleading for interpreting the Figure 11. Actually, our derived results had **higher** proportions of impervious surfaces compared with other previous products according to the figure 11. It should be noted that the x-axis and y-axis represented our GISD30 products and other products, respectively. We could find that almost all the regression lines (black dotted line) were below 1:1 line (black solid line), which clearly demonstrated that the derived results yielded higher proportions of impervious surface compared with all other pervious products. Therefore, the conclusions of Figure 11 and Figure 9 were consistent, namely, the derived products had better ability to capture these fragmented and small impervious surface objects.

[Figure]

Figure 11. The scatter plots between our GISD30 dataset (x axis) and five global 30 m impervious surface products (y axis, GAIA, NUACI, GAUD, GlobeLand30 and GHSL) at the spatial resolution of 0.05°×0.05°. It should be noted that the label of x and y axis was the proportion of impervious surface in each 0.05°×0.05° spatial grid.

5. To justified the outperformance of the derived dataset, the authors should directly compare the accuracies of the derived and previous datasets using the same validation samples, since the accuracies (OA and kappa coefficient) can vary greatly when calculated using different validation samples.

Great thanks for the suggestion. The accuracy metrics between our products and previous datasets have been calculated using same validation dataset in the revised manuscript as:

Except for the consistency analysis, the quantitative accuracy assessments for four global impervious surface products were calculated using the same validation dataset, as listed in the Table 2. The GHSL and GlobeLand30 datasets were excluded because both of them cannot cover the whole period with 5-years interval. Overall, the GISD30 achieved the highest performance with the overall accuracy of 0.901 and kappa coefficient of 0.865, compared with 0.797 and 0.702 for GAIA, 0.843 and 0.748 for GAUD, as well as 0.745 and 0.702 for NUACI. Specifically, in terms of the pervious surfaces, it can be found that all four products achieved similar and great producer's accuracy exceeding 0.94. As the previous comparisons have illustrated that GAIA, NUACI and

GAUD datasets underestimated the impervious surfaces, the user's accuracy of them was lower than the GISD30 dataset. Afterwards, as for the performances of impervious surfaces, the NUACI suffered the lowest user's accuracy and producer's accuracy in 1985 because it only identified the urban areas (Liu et al., 2018) and overestimated some increased impervious surfaces as the early impervious surfaces before 2000 (see Figure 13). Similarly, the GAIA and GAUD also missed some fragmented and small impervious surfaces, so the producer's accuracy of them in 1985 was also greatly lower than that of the GISD30. Then, the accuracy metrics of these increased impervious surfaces were similar to the overall accuracies, namely, the GISD30 could accurate capture the spatiotemporal dynamics of impervious surfaces, followed by the GAUD, GAIA and NUACI datasets.

Table 2. The accuracy metrics of four global 30 m impervious surface dynamic products using the same validation datasets.

| | | P.S. | 1985 | 85~90 | 90~95 | 95~00 | 00~05 | 05~10 | 10~15 | 15~20 | O.A. | Kappa |
|---|---|---|---|---|---|---|---|---|---|---|---|---|
| GISD30 | P.A. | 0.985 | 0.923 | 0.737 | 0.748 | 0.759 | 0.816 | 0.851 | 0.671 | 0.720 | 0.901 | 0.865 |
| | U.A. | 0.958 | 0.935 | 0.809 | 0.779 | 0.848 | 0.863 | 0.837 | 0.855 | 0.882 | | |
| GAIA | P.A. | 0.969 | 0.755 | 0.552 | 0.510 | 0.494 | 0.489 | 0.474 | 0.663 | 0.531 | 0.797 | 0.702 |
| | U.A. | 0.873 | 0.932 | 0.445 | 0.469 | 0.532 | 0.627 | 0.621 | 0.488 | 0.608 | | |
| NUACI | P.A. | 0.940 | 0.660 | 0.459 | 0.348 | 0.317 | 0.422 | 0.395 | 0.482 | | 0.745 | 0.609 |
| | U.A. | 0.839 | 0.796 | 0.160 | 0.348 | 0.398 | 0.624 | 0.626 | 0.608 | | | |
| GAUD | P.A. | 0.978 | 0.855 | 0.516 | 0.554 | 0.528 | 0.551 | 0.520 | 0.571 | | 0.843 | 0.748 |
| | U.A. | 0.896 | 0.901 | 0.535 | 0.620 | 0.642 | 0.693 | 0.637 | 0.614 | | | |

Note: P.S.: pervious surface; 1985: impervious surface before 1985; 85~90: expansion of impervious surface during 1985~1990; …, 15~20: expansion of impervious surface during 2015~2020; U.A.: user's accuracy; P.A.: producer's accuracy; O.A.: overall accuracy.

6. The author should include the recent GAUD product (Liu et al., 2020, cited by the authors) into their comparison, since it provides annual impervious surface layers.

Great thanks for the suggestion. The GAUD product has been added in the cross-comparison section as:

[revised manuscript text omitted]

Figure 11. The scatter plots between our GISD30 dataset (x axis) and five global 30 m impervious surface products (y axis, GAIA, NUACI, GAUD, GlobeLand30 and GHSL) at the spatial resolution of 0.05°×0.05°. It should be noted that the label of x and y axis was the proportion of impervious surface in each 0.05°×0.05° spatial grid.

7. The comparisons presented in figure 12, 13 and 14 are misleading for product GlobeLand30. Many impervious surfaces existed before 1995 are colored with yellow (2000-2005), which is obvious incorrect.

Great thanks for the comment. As the GlobeLand30 only covered the periods of 2000-2020 with the interval of 10 years, it was difficult to identify the impervious surface before 2000 and then we used the yellow color. To avoid the misleading for GlobeLand30, the GAUD data were used to replace the GlobeLand30 in Figure 12-14 as:

[Figure]

[Figure]

Figure 14. The comparisons between GISD30 and four reference datasets (the GAIA products developed by Gong et al. (2020), the NUACI developed by Liu et al. (2018), the GHSL developed by Florczyk et al. (2019), and the GAUD developed by Liu et al. (2020b)) in the two representative arid cities of Phoenix and Johannesburg. In each case, the multi-epoch Landsat imagery, comprised by red, green and blue bands, came from the United States Geological Survey (https://earthexplorer.usgs.gov/).

**Reliability of the assessment**

1. According to the manuscript, the locations of the validation points were randomly generated using the stratified random sampling strategy (L150). In order to better evaluate the results, especially the time-series dynamics of the impervious surface, I suggest a substantial number of validation samples should be placed within (impervious surfaces) and around the fringe of the urban areas (pervious surfaces), which are exactly the most inconsistent and uncertain areas regarding the impervious surface classification problem.

Great thanks for the suggestion. To ensure the **objectivity** of the entire accuracy assessment, the spatial distribution of impervious validation samples must first be guaranteed to be random (Olofsson et al. 2014). If we placed a large number of validation samples within these inconsistent and uncertain areas, it would be unfair for these stable impervious surfaces and pervious surfaces when calculating the accuracy metrics. Then, it should be noted that there was a clear difference for evaluating the global product and regional product. At regional scale, we could artificially place the validation samples and control the size of validation points in these uncertain areas. However, at global scale, we must use the sampling rules to derive the validation samples. In this study, a stratified random sampling based on the proportion of the land-cover areas was adapted to determine the sample size of each land-cover type:

$$n_i = n \times \frac{W_i \times p_i(1-p_i)}{\sum W_i \times p_i(1-p_i)}; \quad n = \frac{(\sum W_i \times \sqrt{S_i(1-S_i)})^2}{[S(\hat{O})]^2 + \sum W_i \times S_i(1-S_i)/N} \approx \left(\frac{\sum W_i S_i}{S(\hat{O})}\right)^2 \quad (1)$$

where $W_i$ was the area proportion for class $i$ over the globe, $S_i$ is the standard deviation of class $i$, $S(\hat{O})$ is the standard error of the estimated overall accuracy, $p_i$ is the expected accuracy of class $i$ and $n_i$ represents the sample size of the class $i$.

Olofsson, P., Foody, G. M., Herold, M., Stehman, S. V., Woodcock, C. E., and Wulder, M. A.: Good practices for estimating area and assessing accuracy of land change, Remote Sensing of Environment, 148, 42-57, https://doi.org/10.1016/j.rse.2014.02.015, 2014.

As we cannot artificially move the place of the validation samples in the uncertain areas, **4682** impervious validation points were further added into the validation dataset under the condition that ensuring the random allocation of validation sample. The spatial distribution of validation samples after adding new impervious samples was revised as:

To quantitatively assess the accuracies of our impervious surface dynamic time-series products, 23,322 validation samples (Figure 2), including 13,236 impervious samples and 9,986 pervious samples, covering the long-term time-series from 1985 to 2020, were randomly generated using the stratified random sampling strategy, and further interpreted on the Google Earth Engine computing platform.

[Figure]

Figure 2. The spatial distribution of the global multitemporal impervious surface validation dataset for 1985-2020.

2. Eyeballing the map of figure 2 there seems to be considerable number of samples located far away from the urban areas. These areas are well detected as pervious surface by many previous methods, or can be easily masked out for example by nightlight observations. Validation samples within these areas do not contribute to justifying the superiority of the derived results, but instead just smoothing the performance difference between the derived and previous datasets.

Great thanks for the comment. Yes, there was a lot of pervious validation samples located away from the urban areas in Figure 2, and these pervious samples would smooth the performance difference between the derived and previous datasets. However, as mentioned in the previous comment, the locations of pervious samples were randomly derived without any artificial participation, meanwhile, to ensure the **objectivity** of the entire accuracy assessment, the pervious sample size was also determined by the stratified random sampling method. If we artificially remove these pervious validation samples, the objectivity of the final accuracy metrics cannot be guaranteed.

3. The number of validation samples in each five-year interval should be large enough to evaluate the impervious surface dynamics during the corresponding time periods. According to the manuscript, the land surface is divided totally 961 geographical tiles (5°×5°) and only 18,540 validation samples were used (L150). I did a rough calculation. There are only ~20 samples on average within each tile, which is definitely too sparse for the evaluation of 30 m classification results in a large area of 5°×5° tile, not to mention that these ~20 samples were divided into eight time periods. Great uncertainties/bias are expected in the calculated OAs and Kappa coefficients with these samples.

Great thanks for the comment. Firstly, we completely agreed that the number of validation samples in each five-year interval should be large enough to evaluate the impervious surface dynamics during the corresponding time periods. However, the collection of validation samples was a very time-consuming and difficult task especially at the early stage (lacking high-resolution imagery), therefore, the previous impervious products (including: GAIA, GAUD, NUACI and GlobeLand30) were validated only using a few thousand points. For example, Gong et al. (2020) used 3500 validation points (including impervious surface and pervious surface) to analyze their performances.

Then, it might be unreasonable to directly distribute the validation samples into geographical tiles, because the spatial distribution of validation samples was non-uniform. In the Figure 2, it can be found that the impervious surface validation samples mainly concentrated on the regions with dense impervious surfaces such as: Europe, North America and China.

Lastly, based on previous suggestions and this comment, **4682** impervious validation points were further added into the validation dataset to comprehensively analyze the performance of the GISD30 dataset. The spatial distribution of validation samples after adding new impervious samples was revised as:

To quantitatively assess the accuracies of our impervious surface dynamic time-series products, 23,332 validation samples (Figure 2), including 13,236 impervious samples and 9,986 pervious samples, covering the long-term time-series from 1985 to 2020, were randomly generated using the stratified random sampling strategy, and further interpreted on the Google Earth Engine computing platform.

[Figure]

Figure 2. The spatial distribution of the global multitemporal impervious surface validation dataset for 1985-2020.

4. As presented in table 1, only hundreds of validation samples were used to evaluate the performances of each five-year period. If I read it correctly, there are less than 1 sample on average within each 5°×5° tile for each five-year period. Assessment with these validation samples is definitely unreliable and cannot truly reflect the quality of the derived dataset. The authors should substantially increase their number of validation samples to achieve a more reliable assessment.

Great thanks for the comment. As mentioned in the previous comments, it might be unreasonable to directly distribute the validation samples into geographical tiles, because the spatial distribution of validation samples was non-uniform. According to the statistics in Results Section, the increased impervious surfaces mainly concentrated on the Asia, Europe and North America, actually, our multitemporal impervious validation samples also mainly distributed around these rapidly expanding areas.

Then, the sample size of these increased impervious surfaces was determined by the stratified random sampling method as:

$$n_i = n \times \frac{W_i \times p_i(1-p_i)}{\sum W_i \times p_i(1-p_i)} \qquad (1)$$

where $W_i$ was the area proportion for class $i$ over the globe, $p_i$ is the expected accuracy of class $i$ and $n_i$ represents the sample size of the class $i$.

Lastly, based on the suggestion and previous comments, we additionally added **4682** impervious validation points into the validation dataset under the condition that ensuring the random allocation of validation sample. Afterwards, the Table 1 has been revised as:

Table 1. The confusion matrix of our global 30 m impervious surface dynamic products using 23,332 validation samples.

| | P.S. | 1985 | 85~90 | 90~95 | 95~00 | 00~05 | 05~10 | 10~15 | 15~20 | Total | U.A. |
|---|---|---|---|---|---|---|---|---|---|---|---|
| P.S. | 9840 | 11 | 20 | 14 | 22 | 21 | 14 | 24 | 20 | 9986 | 0.985 |
| 1985 | 247 | 5408 | 61 | 49 | 41 | 17 | 20 | 8 | 5 | 5856 | 0.923 |
| 85~90 | 28 | 74 | 555 | 27 | 11 | 14 | 19 | 16 | 9 | 753 | 0.737 |
| 90~95 | 43 | 58 | 20 | 556 | 19 | 19 | 10 | 13 | 5 | 743 | 0.748 |
| 95~00 | 70 | 72 | 13 | 31 | 902 | 35 | 31 | 16 | 19 | 1189 | 0.759 |
| 00~05 | 76 | 62 | 12 | 36 | 42 | 1383 | 49 | 29 | 5 | 1694 | 0.816 |
| 05~10 | 52 | 37 | 13 | 14 | 14 | 42 | 1201 | 18 | 21 | 1412 | 0.851 |
| 10~15 | 47 | 52 | 11 | 21 | 23 | 36 | 69 | 566 | 19 | 844 | 0.671 |
| 15~20 | 55 | 59 | 8 | 7 | 14 | 21 | 30 | 43 | 608 | 845 | 0.720 |
| Total | 10268 | 5786 | 686 | 714 | 1064 | 1602 | 1435 | 662 | 689 | 23322 | |
| P.A. | 0.958 | 0.935 | 0.809 | 0.779 | 0.848 | 0.863 | 0.837 | 0.855 | 0.882 | | |
| O.A. | 0.901 | | | | | | | | | | |
| Kappa | 0.865 | | | | | | | | | | |

Note: P.S.: pervious surface; 1985: impervious surface before 1985; 85~90: expansion of impervious surface during 1985~1990; …, 15~20: expansion of impervious surface during 2015~2020; U.A.: user's accuracy; P.A.: producer's accuracy; O.A.: overall accuracy.

5. Apart from point-based validation, the area-based validation strategy, i.e., visual interpreting impervious area in small blocks near the urban fringe and comparing them with the derived results, is more encouraged, considering the sparse distribution of impervious surfaces

Great thanks for the comment. Based on your suggestion, the cross-comparison in the sparse distribution of impervious surfaces has been added in the revised manuscript as:

Lastly, the cross-comparison between GISD30 and four previous datasets in the rural villages (containing sparse impervious surfaces) was illustrated in the Figure 15. Overall, except for our GISD30, the remaining impervious surface datasets failed to identify these small rural buildings around the central villages. In terms of the spatial pattern of villages, the NUACI dataset obviously misclassified a lot of croplands as the increased impervious surfaces and also missed those stable impervious surfaces in the central villages. The GAUD dataset performed well in the early stage and accurately captured these old impervious surfaces, but these increased impervious surfaces after 2000 were missed. In fact, the village experienced significant impervious expansions after 2000 by visually interpreting the multitemporal Landsat imagery. The GAIA partly captured the spatiotemporal expansion in the village, but the impervious areas in the GAIA was obviously smaller than the actual situation, which indicated that the GAIA dataset suffered the underestimation problem in this rural village. Further, it can be found that there was highest consistency between GISD30 and GHSL, both of them captured the expansion pattern of "center-to-periphery", however, the increased impervious surfaces in the GHSL were still less than the actual increases.

[Figure]

Figure 15. The comparisons between GISD30 and four reference datasets (the GAIA products developed by Gong et al. (2020), the NUACI developed by Liu et al. (2018), the GHSL developed by Florczyk et al. (2019), and the GAUD developed by Liu et al. (2020b)) in the rural village. The multi-epoch Landsat imagery, comprised by SWIR1, NIR and red bands, came from the United States Geological Survey.

**Uncertainties**

The uncertainties of the derived impervious surface layers should be discussed but missing in the current manuscript. This dataset at least involved uncertainties from four aspects: 1) the labels of training samples directly collected from the GLC_FCS30-2020 rather than visual interpretation.

Great thanks for the comment. The sensitivity analysis between classification accuracy and the percentage of erroneous samples points has been added in the Discussion section as:

In contrast to supervised classification methods using independent samples for different periods, which require expensive resources to collect multitemporal training samples (Gao et al., 2012; Zhang and Weng, 2016), we used prior global land-cover products and the spectral generalization strategy to automatically monitor the impervious surface dynamics. Firstly, as the reliability of the training samples was demonstrated to directly affect the final classification accuracy, we combined the impervious layers in the GLC_FCS30-2020 and GlobeLand30-2020 land-cover products to derive candidate impervious training samples, and then adopted the spatial homogeneity filtering to further ensure the reliability of each sample in 2020. In order to assess the accuracy of training samples, we randomly selected 10,000 impervious surface samples from the global sample pool, and the 10,000 random samples were interpreted by visual interpretation. The validation result showed that these impervious training samples achieved an overall accuracy of 95.52% in 2020. To demonstrate whether the erroneous training samples can affect the performance of the classifiers, we gradually increased the percentage of erroneous training samples with the step of 1 % and then repeated 100 times illustrated in Figure 16, it can be found that the local adaptive random forest models had great performance to be resistant to noise and erroneous training samples, and the overall accuracy and impervious surface producer's accuracy kelp stable when the percentage of erroneous training samples were controlled within 40% and then obviously decreased after exceeding the threshold. Similarly, Gong et al. (2019b) also found that the overall accuracy kept stable when the percentage of erroneous training samples was within 20%. Therefore, the training samples derived in Section 3.1 were accurate enough for monitoring impervious surface dynamics.

[Figure]

Figure 16. The relationship between overall accuracy and impervious producer's accuracy with the percentage of erroneous training samples using the random forest classification model.

2) The migration of reflectance spectra of impervious surfaces measured in 2020 to other periods.

Great thanks for the comment. The feasibility of generalizing reflectance spectra of impervious surfaces measured in 2020 to other periods has been added in the Discussion Section as:

Furthermore, many studies have demonstrated that the spectral inconsistency between migrated spectra and classified imagery directly affects classification accuracy (Woodcock et al., 2001; Zhang et al., 2018). In this study, we used continuous Landsat imagery to preclude the effects of different sensors, and adopted a seasonally composited method with relative radiometric normalization to minimize the influence of temporal difference. **We toke the Yangtze River Delta as an example to draw scatterplots for NIR reflectance of impervious surfaces in 2020 against other periods at the growing season after relative radiometric normalization**

**illustrated in Figure 17. There were significant consistency in NIR band between reference period and other periods and most scatters were distributed on both sides of the regression line. In terms of the regression slope, the slope got closer and closer to 1.0 as time increased, which mainly caused by the shorter temporal difference and denser Landsat imagery at later periods. According to the distribution of scatter points and the regression lines, there was no systematic bias between reference data and other data, which also demonstrated that it was feasible to generalize the reflectance spectra of impervious surfaces in the 2020 to other periods.**

[Figure]

Figure 17. The scatterplots for NIR reflectance of impervious surface in 2020 (y-axis) against other periods (x-axis) after relative radiometric normalization in the Yangtze River Delta region.

3) The reliability of the random forest classifiers for each tile and time period.

Great thanks for the comment. The reliability of the random forest classifiers directly depended on the **confidence of training samples** and **the spectral consistency for impervious surface reflectance spectra** between reference period and other periods. The previous two comments have quantitatively demonstrated the derived training samples were accurate enough for monitoring impervious surface dynamics, and the spectral consistency ensure the feasibility of generalizing the reflectance spectra of impervious surfaces in the 2020 to other periods. Therefore, after discussing the **confidence of training samples** and **the spectral consistency**, the local random forest classifiers used by the study were reliable for monitoring time-series impervious surface dynamics.

4) The temporal consistency correction used to smooth the independent classification results. It is not clear how these uncertainties propagate along the entire derivation procedure and to what extent these uncertainties contribute to the final derived dataset.

Great thanks for the comment. The temporal consistency checking is an optimization algorithm for time-series land-cover monitoring (eliminating the "salt-pepper" noisy in the multi-epoch impervious surface maps and using the irreversible assumption to remove the illogical transitions), the discussion about the method has been added in the Discussion as:

Lastly, to optimize the time-series impervious maps and minimize the influence of classification error, the temporal consistency checking post-processing method proposed by the Li et al. (2015) was adopted. It mainly used the spatiotemporal correlation information to eliminate the "salt-pepper" noisy in the multi-epoch impervious surface maps, and used the irreversible assumption to remove the illogical transitions. Li et al. (2015) quantitatively demonstrated that the post-processing method improved the overall accuracy by about 6% for monitoring impervious dynamics in Beijing, China. Recently, this post-processing method was involved for producing GAIA dataset (Gong et al., 2020) and optimizing time-series land-cover maps in China (Yang and Huang, 2021), both of them demonstrated that temporal consistency checking improved the reliability and consistency of the classification results by integrating the spatio-temporal context information.

**Others**

1. There are many typos in the manuscript. To name only a few for example from the methodology section: brackets in L220, L223, L239, L271.

Great thanks for pointing out the issues. The manuscript has been totally revised and then we will invite a professional team to carefully polish the revised manuscript again.

2. Some paragraphs only consist one or two sentences, e.g., L178-183, L191-196, L214-218.

Great thanks for the comment. These broken paragraphs have been merged in the revised manuscript.

3. Although the manuscript is readable, there are still many inappropriate expressions and grammar errors throughout the entire manuscript. Please consult a native English speaker or a commercial proofreading service.

Great thanks for the suggestion. The manuscript has been polished before summiting. At the same time, after the article is received, we will also invite a professional team to carefully polish the article again.

---

## Author Comment (AC2)

**Response to comments**

**Paper #:** essd-2021-285

**Title:** GISD30: global 30-m impervious surface dynamic dataset from 1985 to 2020 using time-series Landsat imagery on the Google Earth Engine platform

**Journal**: Earth System Science Data

**Reviewer #2**

In this manuscript, the authors produced a global 30 m impervious surface dynamic dataset from 1985 to 2020 using the spectral generalization method and time-series Landsat imagery on GEE, and cross-compared the dataset with four existing global 30 m impervious surface products. The manuscript is well arranged, and the logic is clear. Even so, there are still some modifications need to be finished before it accepted. The following are the questions and some mistakes in this manuscript.

Great thanks for the positive and careful comments. The manuscript has been improved according to your and another reviewer's comments.

Line 130: What is the size of the areas where these data are missing? Whether the assumption that their land cover types remain unchanged will affect the accuracy of the final classification results.

Great thanks for the comment. According to our statistics, the proportions of missing Landsat observations in the first three periods (before 1985, 1986-1990 and 1991-1995) were 37.3%, 11.3% and 11.4%, respectively. The missing Landsat observations in the second and third periods (1986-1990 and 1991-1995) mainly concentrated on the Northeast Asia in which contained a small number of impervious surfaces, so the unchanged assumption had little effect in these two epochs. As for the first epoch, the missing observation areas covered the East Asia and the whole Oceania continent, so the assumption would affect the accuracy of final results. Our manuscript in Section 4.2 of accuracy assessment has also illustrated:

"However, the user's accuracy for the expansion of impervious surface after 2000 was higher than that before 2000, which was mainly affected by the sparser available Landsat observations before 2000 in Figure 1. Similarly, Gong et al. (2020) also found that the monitoring uncertainty before 2000 was greater than after 2000."

It should be noted that almost all time-series impervious surface products (including: GAIA, NUACI, GUD and GHSL) also used the unchanged assumption to monitor impervious surfaces in these missing Landsat observation areas.

Line 132: What do these numbers in the legend of Figure 1 mean? Do they represent the number of scenes in the images from different years? Why do not marked in the legend of the figure?

Great thanks for the comment. Yes, the numbers in the lower right of Figure 1 represented the available Landsat imagery from different years. Based on the suggestion, the figure has been revised as:

[Figure]

Figure 1. The spatial distributions of the available Landsat observations from 1985 to 2020, with 5-year intervals.

Lines 161~165: I do not understand this part of the text. What do you mean like 'the location of each validation sample in rural areas was moved to the center of the impervious object' With such a large sample set, how did you identify the validation sample in rural areas?

Great thanks for the comment. I am sorry that it was a litter confusing in explaining the collection of impervious surface validation samples. Actually, only a small amount of impervious validation samples in the rural areas was moved to the center of impervious objects for minimizing the effect of geometry error between the high-resolution imagery in Google earth and the Landsat imagery, because we found some rural impervious validation points in the boundary of impervious objects actually belonged to the pervious surfaces after projecting to the Landsat imagery. There was a total of 649 rural impervious surface samples have been moved according to our statistics. The part has been revised as:

"as the spatial heterogeneity of the impervious surface was usually higher than that of natural land-cover types, the impervious area in a 30 × 30 m window should comprise more than 50% when identifying impervious samples (Zhang et al., 2020). **Meanwhile, to minimize the effect of geometry registration between validation samples and our products, the geolocations of these rural impervious surface samples, located in the transition areas between the impervious objects (such as buildings and roads) and pervious objects, were re-positioned in the center of the objects.**"

The rules of how we identify the validation sample in rural areas were: 1) the size of impervious surface blocks (the rural areas were usually fragmented and smaller than the cities); 2) the land-cover distributions around the validation sample (the surrounding environment in the rural areas were usually the cropland, forest and grassland).

Lines 356~357: 'we categorized the time-series impervious surface dynamic into 9 independent strata, including: pervious surfaces, impervious surfaces before 1985, and expanded impervious surfaces during 1990-1995, 1995-2000, 2000-2005, 2005-2010, and 2015-2020.' Whether 1985-1990 and 2010-2015 are missing from the presentation.

Great thanks for pointing out the mistake. Yes, the 1985-1990 and 2010-2015 are missing in our presentation. It has been added in the revised manuscript as:

"as opposed to traditional period-by-period accuracy assessments, we categorized the time-series impervious surface dynamic into 9 independent strata, including: pervious surfaces, impervious surfaces before 1985, and expanded impervious surfaces during **1985-1990**, 1990-1995, 1995-2000, 2000-2005, 2005-2010, **2010-2015** and 2015-2020. We then calculated a comprehensive confusion matrix for these nine strata."

Line 364: 'Further, we selected three types of cities (mega-cities, tropical cities and arid cities)…' Why choose these three types of cities to reveal the spatiotemporal dynamic.

Great thanks for the comment. The reasons why we choose three types of cities have been added in the manuscript as:

"we compared the time-series impervious areas of five products in six continents, and further analyzed the spatial consistency between GISD30 and five comparative datasets at the global scale. Further, we selected three types of cities (mega-cities, tropical cities and arid cities) and one rural area to illustrate the performance of five global 30 m impervious surface products used for capturing the spatiotemporal dynamic. **The reasons why we chose these types of cities and rural areas were that (1) the mega-cities usually experienced more intense urbanization, we could more intuitively understand whether there were commission error and omission error in each product; (2) the tropical cities usually mean sparser observations caused by the cloud coverage, so we could analyze the stability and robustness of each product in the tropical cities; (3) the arid cities were selected to analyze the ability of each product to distinguish between impervious surfaces and similar land types (arid soils); (4) the rural area contained sparse impervious surfaces and were prone to suffer the underestimation problem.**"

English writing needs to be further improved; some sentences are too long to affect the understanding of the article. The sentences can be broken down. e.g.: Lines 123~127, 127~131…

Great thanks for the comment. Based on your suggestion, the manuscript has been totally revised and then we will invite a professional team to carefully polish the revised manuscript again.

Line 22: '… similar to in …' should be 'similar to'?

Thanks for the comment. It has been revised and we have invited a professional team to carefully polish the revised manuscript.

Line 146: '…normalized difference water index (NDWI) and NDWI…'should be 'NDVI'?

Great thanks for pointing out this mistake. It was corrected in the revised manuscript.

---

## Author Comment (AC3)

**Response to comments**

**Paper #:** essd-2021-285
**Title:** GISD30: global 30-m impervious surface dynamic dataset from 1985 to 2020 using time-series Landsat imagery on the Google Earth Engine platform
**Journal**: Earth System Science Data

**Community Comment #1**

Thanks for your contributions! I have several questions:
Great thanks for your comments. Each comment has been responded at below.

1. What projection is used for the generated products? Did you equal area to facilitate calculation of areas? If not equal area, which approach did you use to account for discrepancies in area of pixels?
Great thanks for the comment. The GISD30 dataset used the geographical projection for the convenience of users. Yes, when we calculated the impervious surface area at regional and continent scales, the dataset was reprojected to the sinusoidal equal area projection because there was serious **area deformation in the latitude direction** for geographical projection.

2. What approach did you use for calculating the area: directly from the map or from samples? The former is a pixel counting estimator which is biased. From confusion matrices, one can observe that PA and UA are not balanced, and therefore land cover area over/under-estimation can happen. For all area calculations, what are the uncertainties, since produced maps are not error-free?
Great thanks for the comment. All area statistics in the manuscript were derived from the map. Yes, based on the confusion metrics, the area statistics in current manuscript were biased. According to the suggestion, the uncertainties of the area calculations have been calculated.
Specifically, based on the work of Olofsson et al. (2013), the area uncertainties can be estimated as:

$$Uncertain = \pm 2 \times A \times \sqrt{\sum_{i=1}^{q} W_i^2 \frac{\frac{n_{ij}}{n_{i.}}\left(1-\frac{n_{ij}}{n_{i.}}\right)}{n_{i.}-1}}, \quad W_i = A_{imp}/A$$

where $A_{imp}$ and A is the mapped impervious area in this study and the total area of the map, $q$ is the number of land-cover types, $n_{ij}$ and $n_{i.}$ denotes the element in the confusion matrix.

Based on the confusion matrix in the Table 1 and the mapped impervious areas, the area uncertainties of impervious surfaces were calculated in 1985 and 2020 were: ± **4.502×10⁴ km² in 1985** and ± **4.455×10⁴ km² in 2020**, respectively.

Olofsson, Pontus, et al. "Making better use of accuracy data in land change studies: Estimating accuracy and area and quantifying uncertainty using stratified estimation." Remote Sensing of Environment 129 (2013): 122-131.